# RNA binding of Hfq monomers promotes RelA-mediated hexamerization in a limiting Hfq environment

Pallabi Basu [1], Maya Elgrably-Weiss [1], Fouad Hassouna[2], Manoj Kumar[2], Reuven Wiener[2] & Shoshy Altuvia[1✉]

The RNA chaperone Hfq, acting as a hexamer, is a known mediator of post-transcriptional regulation, expediting basepairing between small RNAs (sRNAs) and their target mRNAs. However, the intricate details associated with Hfq-RNA biogenesis are still unclear. Previously, we reported that the stringent response regulator, RelA, is a functional partner of Hfq that facilitates Hfq-mediated sRNA–mRNA regulation in vivo and induces Hfq hexamerization in vitro. Here we show that RelA-mediated Hfq hexamerization requires an initial binding of RNA, preferably sRNA to Hfq monomers. By interacting with a Shine–Dalgarno-like sequence (GGAG) in the sRNA, RelA stabilizes the initially unstable complex of RNA bound-Hfq monomer, enabling the attachment of more Hfq subunits to form a functional hexamer. Overall, our study showing that RNA binding to Hfq monomers is at the heart of RelA-mediated Hfq hexamerization, challenges the previous concept that only Hfq hexamers can bind RNA.

[1] Department of Microbiology and Molecular Genetics, IMRIC, The Hebrew University, Hadassah Medical School, Jerusalem, Israel. [2] Department of Biochemistry and Molecular Biology, IMRIC, The Hebrew University, Hadassah Medical School, Jerusalem, Israel. ✉email: shoshy.altuvia@mail.huji.ac.il

As a rule, most RNA-based regulation involves the function of RNA binding proteins including Hfq, ProQ, cold shock proteins, and proteins of the CsrA family[1–8]. Out of which, Hfq and its associated small regulatory RNAs were acknowledged as significant key players of a large network of post-transcriptional control of gene expression in Gram-negative bacteria. Acting as an RNA chaperone, Hfq facilitates basepairing between small regulatory RNAs and their target mRNAs, thereby leading to altered stability and/or translation of the target genes[9–12]. The importance of Hfq for global RNA regulation has been substantiated through studies showing that Hfq interacts with a great number of different sRNAs and mRNAs species[1,2,13]. Hfq was also shown to bind rRNAs and tRNAs, suggesting an effect on ribosome biogenesis and translation efficiency implicating Hfq as a global regulator[14].

Hfq structural studies showed that the protein forms a doughnut shaped homo-hexamer. The hexameric ring reveals four sites that can interact with RNA: proximal and rim faces interact with uridines present in the 3′ end of sRNAs, distal face interacts with ARN motifs present in the target mRNAs and C-terminal tail ensures the release of the RNAs from Hfq, enabling Hfq recycling[15–17]. In addition to its affinity for RNA, Hfq interacts with components of the RNA decay machinery such as poly(A) polymerase, polynucleotide phosphorylase, RNase E, and the transcription termination factor Rho[18–22].

In vitro studies indicated that Hfq transitions from monomer to hexamer at about 1 µM of Hfq protein and that RNA-bound Hfq hexamer is a stable complex[23]. At higher concentrations, Hfq predominantly forms multimers, whereas upon dilution, the subunits dissociate, indicating that multimerization depends on the Hfq micro-environment and that the interactions are reversible[23]. Mutations in Hfq that impair RNA binding either strongly destabilize the hexamer or prevent hexamer association to multimers, indicating that RNA binding is coupled to hexamer assembly[24–26]. Whether RNA binding coincides with hexamerization which requires initial disassembly of Hfq, assuming that RNA can bind individual Hfq subunits to form a new RNA-bound complex or whether hexamers are the only forms capable of RNA binding which necessitates random recycling of new RNAs on the surface of Hfq are some of the unresolved issues regarding the Hfq-RNA biogenesis. Both the options also raise the possibility that other regulators chaperones Hfq-RNA biogenesis.

While investigating expression regulation by RyhB sRNA, we discovered that the stringent response regulator protein RelA is a functional partner of Hfq mediating RyhB target regulation[27]. We suggested that RelA impacts RyhB target mRNA regulation by promoting assembly of Hfq monomers into hexamers and thereby enabling low and ineffective concentrations of Hfq to bind RNA[27].

The RelA protein of *Escherichia coli* is a ribosome-dependent (p)ppGpp synthetase that is activated under conditions of amino acid starvation[28,29]. Once produced, (p)ppGpp modifies the activities of multiple cellular targets, including enzymes for DNA replication, transcription, translation, ribosome assembly, cellular metabolism, and genome stability[28,30–32]. RelA synthetase activity resides within the amino terminus of the protein whereas the carboxy terminus enables regulation of the synthetase function in a ribosome-dependent manner[33,34].

Here, we show RNA binds Hfq monomers and that RelA by interacting with a specific sequence in the sRNA, stabilizes the initially unstable complex of RNA-Hfq monomer, promoting the association of additional Hfq subunits to form the hexameric complex. Overall, our study challenges the previous concept that only Hfq hexamer can bind RNA and introduces a new chaperone-like regulator that mediates RNA-bound Hfq hexamerization.

## Results

**RelA amino terminus facilitates repression of RyhB targets by RyhB.** To identify domains in RelA that promote Hfq activity, we carried out deletion mapping in which either the N-terminus or the C-terminus of RelA were eliminated. The genetic system used to test these constructs included RyhB target reporters (*sdhC-lacZ* and *sodA-lacZ*) as single copies, chromosomally encoded Hfq, RyhB expressing plasmids and P15A moderate copy plasmids encoding RelA in Δ*relA*Δ*ryhB* strain. RelA and truncated RelA carrying only the N-terminal domain (pRelA-ΔCTD) enabled repression of RyhB targets by RyhB (Fig. 1a and Supplementary Fig. 1). In contrast, RelA-ΔNTD, carrying only the C-terminal domain of RelA failed to enable repression, indicating that the N-terminal domain of RelA is essential for RyhB-mediated regulation of *sdhC* and *sodA*. We used the same genetic system to isolate RelA mutants by subjecting the RelA gene to random mutagenesis. Two single-point mutations in RelA, C289Y, and T298I, that reduced the repression of RyhB target genes were clustered in one helix of the RelA N-terminal domain (Fig. 1a and Supplementary Figs. 1 and 2).

To test whether RelA-mediated regulation is sequence-specific or resulting from structural elements; we changed the cysteine residue (small and non-polar) at position 289 to alanine harboring characteristics similar to cysteine. RelA:C289A rescued 50% of *sodA-lacZ* repression and 85% of *sdhC-lacZ* repression by RyhB (Fig. 1a and Supplementary Fig. 1). As wild type RelA carries tyrosine, an aromatic amino acid residue at position 290, the mutational change C289Y resulted in two consecutive tyrosine residues that were expected to cause steric hindrance because of their bulky side chains[35]. The double mutant RelA:C289Y;Y290C in which the tyrosine residue at position 290 was changed to cysteine was more effective in mediating repression than the single C289Y mutant, suggesting that C289Y causes steric hindrance and that RelA-mediated regulation relies primarily on structural elements (Fig. 1a and Supplementary Fig. 1).

As RelA mutants affecting repression of RyhB targets reside in the amino terminus of RelA, we examined whether (p)ppGpp production correlated with the RelA regulatory activity of basepairing RNAs. The double mutant strain Δ*relA*Δ*spoT* fails to grow in M9 minimal medium unless supplemented with a plasmid producing (p)ppGpp. Δ*relA*Δ*spoT* cells carrying the empty vector plasmid and pRelA-ΔNTD did not grow on minimal plates, whereas the growth of Δ*relA*Δ*spoT* strains carrying the RelA mutants unable to facilitate RyhB target repression (pRelA:C289Y, pRelA:T298I) and those supporting repression by RyhB (pRelA:C289A, pRelA:C289Y;Y290C) was comparable to cells expressing wild type RelA, indicative of (p)ppGpp production (Supplementary Fig. 3). In addition, we constructed RelA:Q264E[36], a (p)ppGpp synthetase deficient RelA mutant (Supplementary Fig. 3). RelA:Q264E enabled repression of *sodA-lacZ* and *sdhC-lacZ* by RyhB (Fig. 1a and Supplementary Fig. 1), further indicating that (p)ppGpp production and RelA regulation of basepairing RNAs are distinct functions, although both reside in the N-terminal domain.

**RelA amino terminus induces Hfq assembly.** Previously, we have shown that purified wild type RelA enhanced the RNA binding activity of Hfq and Hfq oligomerization in vitro[27]. The in vivo phenotype of RelA mutants prompted us to examine their effect on Hfq RNA binding and on Hfq quaternary structure. Gel mobility shift assays showed that low concentrations of Hfq (5 nM) were insufficient to bind *sodA* RNA unless incubated in the presence of RelA (Fig. 1b). The in vivo inactive mutant RelA:C289Y failed to facilitate binding of RNA by Hfq, whereas the

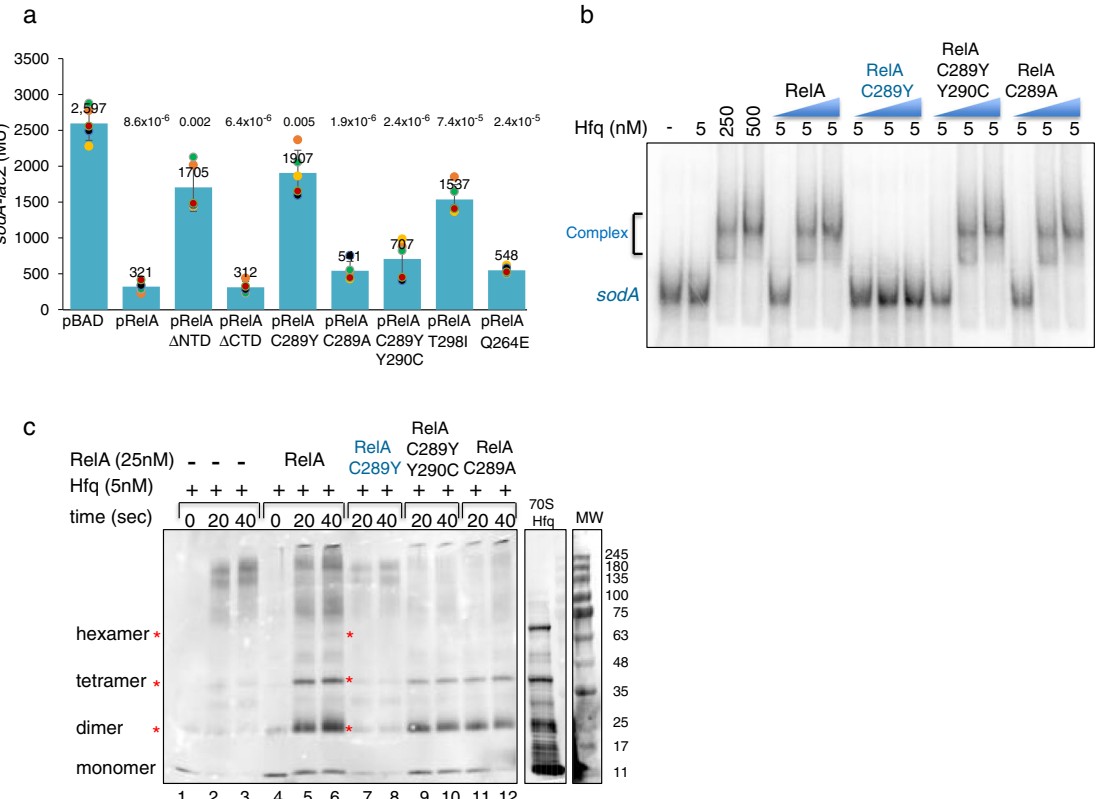

**Fig. 1 RelA amino terminus domain induces Hfq assembly. a** β-galactosidase assay to determine the effect of plasmids encoded RelA alleles on repression of *sodA-lacZ* target gene fusion by RyhB. Expression of RelA from BAD promoter was induced with (0.2%) arabinose. $n = 5$ biological independent samples were examined. Two-tailed unpaired *t*-test were performed; the absolute *p* values are indicated. Data are presented as mean values ± SD. **b** RelA facilitates binding of RNA to Hfq in vitro (EMSA). Gel mobility shift assay of radiolabeled *sodA* RNA (210 nt; 1 nM) incubated with Hfq without and with 50, 250, and 500 nM of purified wild type or RelA mutant proteins as indicated (blue triangle). Incubations were carried out at 22 °C for 10 min and the products were separated by 4% native gel electrophoresis. Unbound (*sodA*) and bound (complex) RNA is indicated in the figure. **c** RelA enhances the multimerization of Hfq protein (western). Hfq was incubated with or without purified RelA proteins for 10 min at 22 °C and samples for zero time point was collected. Thereafter, the products were crosslinked with 0.2% glutaraldehyde at 22 °C. Samples were collected at the indicated time points and the reactions were stopped with 200 mM fresh glycine. The proteins separated in 4–20% MOPS gradient gels (GenScript ExpressPlus™) were detected using α Hfq antibody. Asterisk indicate the Hfq multimers. BLUeye Prestained Protein ladder (MW) and 70S ribosomes (4 μg 70S Hfq[27]) prepared from wild type cells were loaded as markers to visualize various multimers of Hfq protein. Source data provided as a source data file.

active suppressor mutants; RelA:C289A and RelA:C289Y;Y290C enhanced the binding activity of Hfq similar to wild type RelA (Fig. 1b).

To evaluate the effect of RelA mutants on Hfq quaternary structure, Hfq protein incubated with RyhB sRNA and with or without RelA was exposed to glutaraldehyde, a protein cross-linking reagent. The reaction products were separated by SDS-PAGE and detected using α Hfq antibody. The pattern of Hfq oligomerization obtained upon incubation with RelA:C289A and RelA:C289Y;Y290C was similar to that detected with wild type RelA (note the presence of dimers, tetramers, and hexamers), whereas the pattern of Hfq oligomerization obtained with RelA: C289Y was similar to the pattern detected with Hfq alone (Fig. 1c). Combining the in vivo and the in vitro results indicates that RelA supports expression regulation by basepairing RNAs and enhances Hfq RNA binding by facilitating oligomerization of Hfq. Furthermore, the function that supports basepairing RNAs is distinct from the synthetase activity although both reside in RelA amino terminus.

**RelA binds RNA bound by Hfq.** The functional interaction between RelA and Hfq motivated us to investigate the in vivo molecular interaction between these two proteins. Co-

immunoprecipitation using α RelA antibody showed that Hfq precipitated in a complex with RelA and identified RNA as the mediator connecting between RelA and Hfq (Fig. 2a). Hfq did not precipitate with RelA when the lysate was treated with RNase A, suggesting that in the absence of RNA the complex disassembles. Likewise, Hfq did not precipitate with RelA:C289Y mutant that is unable to promote RNA binding by low concentrations of Hfq or induce Hfq assembly. These results demonstrate that in vivo, RNA links between RelA and Hfq forming a complex.

To visualize direct binding between RelA and RNA, in vitro labeled *sodA* or RyhB RNAs were incubated with purified wild type RelA and RelA:C289Y mutant followed by UV crosslinking. Thereafter, the unbound and thus unprotected RNA residues were subjected to degradation by RNase A or left intact. Proteins covalently bound to untrimmed, labeled RNA (Fig. 2b) or to trimmed RNA (Supplementary Fig. 4) were then detected in SDS gels. The results demonstrate that wild type RelA binds both RNAs, however, RyhB binding by RelA is much stronger when compared to RelA binding affinity for *sodA*. The addition of unlabeled competitor RNAs eliminated binding of the labeled RNAs and RelA:C289Y showed no binding, indicating RelA: C289Y that is unable to induce Hfq assembly is also incapable of RNA binding (Fig. 2b and Supplementary Fig. 4). The decrease in binding detected with 1 μM of RelA (Fig. 2b) is probably due to

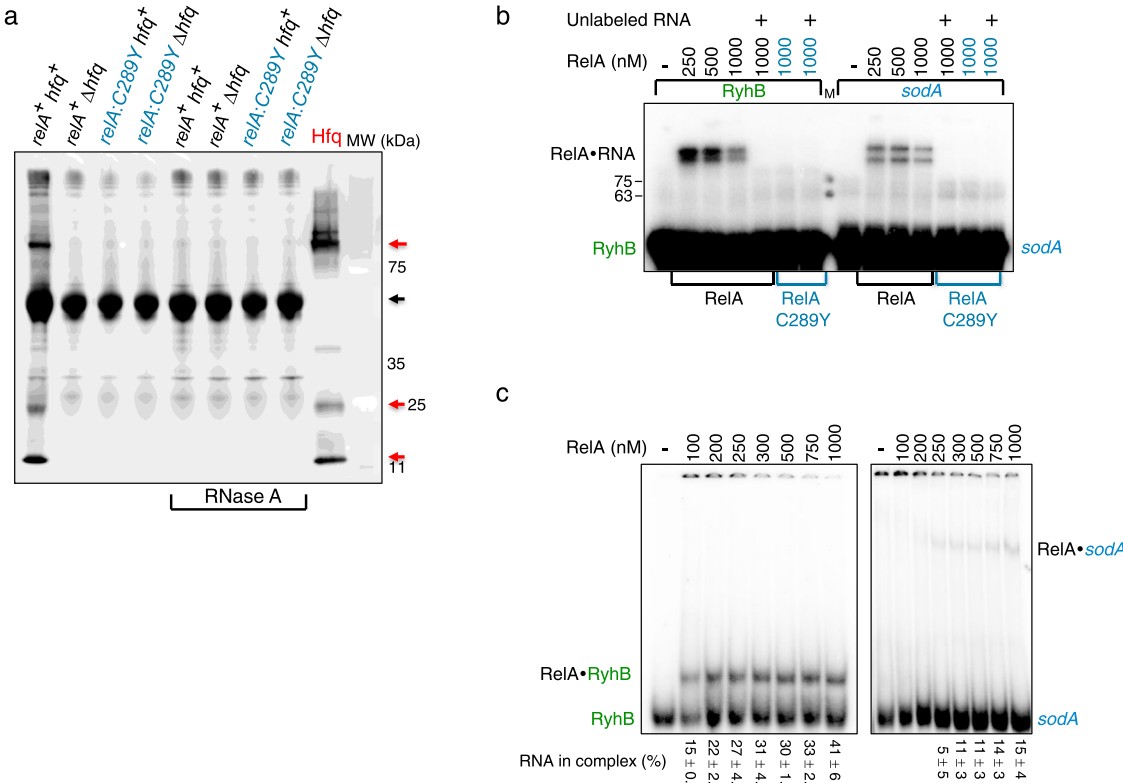

**Fig. 2 RelA binds RNA. a** Co-immunoprecipitation carried out with α RelA antibody and cell lysates of wild type RelA (black) and RelA:C289Y mutant (blue). The lysates were treated with RNase A (100 µg/ml) or left untreated as indicated. Hfq was detected by western using α Hfq antibody. Purified Hfq (100 nM) was used as control. Red arrows indicate different forms of Hfq, while black arrow indicates the heavy chain of α RelA antibody. BLUeye Prestained Protein ladder (MW). **b** In vitro binding of RyhB and *sodA* by RelA. Wild type RelA (black) or RelA:C289Y (blue) incubated with labeled RNAs (1 nM; RyhB 90 nt; *sodA* 98 nt) were UV crosslinked. Competitor unlabeled RNA (100 nM) was added to the reaction mixtures as indicated. The binding products were analyzed by 15% SDS-PAGE. The estimated MW of the RNA•RelA complex is 107–113 kDa. BLUeye Prestained Protein ladder (MW). **c** Gel mobility shift assay of (1 nM) radiolabeled *sodA* (56 nt) or RyhB (50 nt) RNAs incubated with increasing concentrations of RelA as indicated. Incubations were carried out at 22 °C for 10 min. The products were UV crosslinked before loading on 4% native gel electrophoresis. RNA in complex (%) was calculated from four different experiments for each RNA. Source data provided as a source data file.

formation of high molecular weight RelA complexes incapable of RNA binding. These complexes are likely formed at high concentrations of RelA by UV-mediated increased covalent crosslinking of aromatic residues[37].

The higher affinity of RelA for RyhB was further confirmed by gel mobility shift experiments. Data presented in Fig. 2c show that under these conditions RelA is capable of binding up to 30–40% of the RyhB RNA, whereas the binding affinity of RelA to *sodA* mRNA is much weaker. As the in vivo complex of RNA bound by Hfq and RelA is sufficiently stable to be precipitated by α RelA antibody, yet in vitro RNA binding by RelA is limited, we suspect that RelA binding of RNA that is structurally modified by Hfq is more efficient. In the absence of Hfq, the interaction of RelA with unaltered RNA is more elusive.

**RelA binds RNA with a specific sequence.** To define domains in *sodA* and RyhB RNA that interact with RelA, we mapped the sites protected by RelA using dimethyl sulfate (DMS) that methylates unpaired adenosine and cytidine residues or RNase T1 that is specific for unpaired guanosine residues. The modified nucleotides and cleavage sites were mapped by primer extension. The results displayed in Fig. 3a–c and summarized in Fig. 3d show that RelA protects the sequence GGAGA in both *sodA* and RyhB. RyhB also consists of a variation of this sequence (GGAAGA) but RelA did not protect this site. The pattern of RNA probing upon incubation with RelA:C289Y mutant was similar to the pattern

obtained in the absence of RelA, further confirming that C289Y mutant does not bind RNA. To confirm that GGAGA is the site RelA interacts with, we changed this sequence to ACUCU in *sodA* (*sodAm*) (Fig. 3a, b, d). The pattern of RNA probing of *sodAm* incubated with wild type RelA or RelA:C289Y mutant was identical to the pattern detected in the absence of RelA indicating that RelA binds the sequence GGAGA which intriguingly resembles the ribosome binding Shine–Dalgarno (SD) sequence.

**RelA induces Hfq assembly by binding RNA with GGAGA.** To further confirm that RelA-mediated Hfq assembly requires interaction with GGAGA, we investigated the assembly pattern of Hfq in the presence of *sodA* and RyhB wild type and mutant RNAs. To this end, we constructed *sodA* that lacks the GGAGA region (ΔSD), as well as *sodAm* and RyhBm in which GGAGA was changed to ACUCU and CAUCU, respectively. In RyhBm, the variant site GGAAGA was also mutated to GGUUCA (see sequence Supplementary Fig. 5). Protein crosslinking showed that in the absence of RelA, the addition of either of the RNAs had no effect on assembly of low concentrations of Hfq (Fig. 4a, c lanes 1–9 and quantitation in Supplementary Fig. 6a, c). In the presence of RelA, the addition of wild type *sodA* or RyhB resulted in increased levels of Hfq dimers, tetramers and pentamers (Fig. 4b, d; lanes 1–6 and Supplementary Fig 6b, d), whereas the assembly pattern of Hfq presented with *sodA*-ΔSD, *sodAm* or with RyhBm was similar to that detected without any RNA (Fig. 4b, d and Supplementary Fig. 7

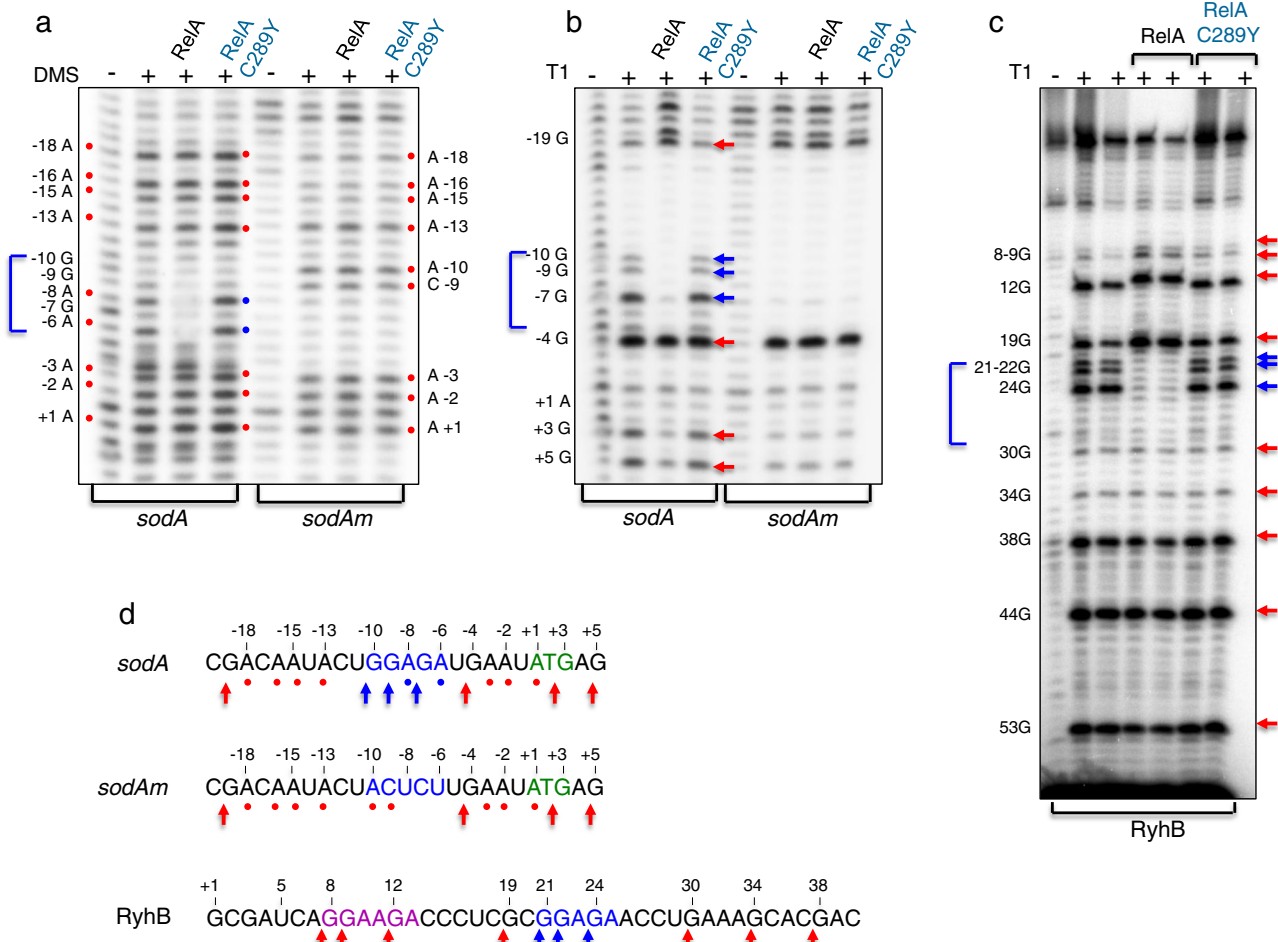

**Fig. 3 RelA interacts with RNA through a specific sequence. a** Footprinting of RelA using DMS modification. Wild type (black) and mutant (blue) RelA proteins (5 pmol) incubated with (0.5 pmol) RNAs (RyhB 90 nt; *sodA* 98 nt) were exposed to DMS modification (0.3%) for 5 min at 25 °C. *sodA* carries an intact GGAGA sequence while *sodA*m carries ACUCU. Reverse transcription of untreated (−) and DMS treated (+) RNA samples. The red circles indicate the positions methylated by DMS. The numbers on the right and left indicate the sequence position relative to the nucleotide A of the start codon of *sodA* (+1). Wild type RelA protects residues A-8 and A-6 from methylation (blue circles). Nucleotides A-10 and C-9 in *sodA*m RNA are methylated (red circles) in the presence of either wild type or RelA:C289Y mutant. **b**, **c** Footprinting of RelA using RNase T1. RNAs and proteins incubated as in **a** were treated by RNase T1 (0.1 U) for 5 min at 37 °C (**b**) or with 0.2 U and 0.4 U (**c**). Reverse transcription of untreated (−) and RNase T1 treated (+) RNA samples. The numbers on the left indicate the sequence position relative to the nucleotide A of the start codon of *sodA* (**b**) and the transcription start site +1 of RyhB (**c**). The red arrows indicate the positions of the G residues cleaved by RNase T1, while the blue arrows represent the regions of protection. **d** RelA protects GGAGA sequence of RyhB and *sodA*. The sequences GGAGA (blue), AUG (green), and variant GGAAGA (purple) are denoted. In *sodA*m GGAGA was changed to ACUCU. Red circles and arrows indicate strong modification and cleavage sites. Blue circles and arrows indicate the region protected by RelA. The products were analyzed in 6% acrylamide 8 M urea-sequencing gel. Source data with MW labeled marker (pUC18) and sequencing reactions is provided as a source data file.

lanes 1–3 and 7–9). Taken together, the results indicate that in the absence of RelA, RNA has no effect on oligomerization of low concentrations of Hfq. However, RNA plays a significant role in RelA-induced Hfq hexamerization that is driven by RelA interacting with RNA carrying a GGAGA site.

**RelA stabilizes complexes of RNA-bound Hfq monomers to form hexamers.** To follow the steps of RelA-induced Hfq assembly to hexamers, Hfq (5 nM) and labeled RyhB RNA, with or without RelA were crosslinked and the products were separated on SDS gels. Intriguingly, we detected binding of labeled RyhB to one Hfq monomer (Fig. 5a). The binding was visible only in the presence of RelA, unlabeled RNA competed with the labeled one for Hfq binding and reactions carrying RelA:C289Y mutant showed no binding, indicating that RelA stabilizes

complexes of RNA associated with Hfq monomers by interacting with the RNA. Similarly, incubation of labeled *sodA* with Hfq in the presence of RelA resulted in formation of *sodA*•Hfq complex. However, the complex *sodA*•Hfq was significantly weaker compared to RyhB•Hfq, suggesting that sRNA is a much better substrate for RelA (Supplementary Fig. 8). Incubation of labeled *sodAm* RNA with RelA and Hfq resulted in no binding, indicating the preference of RelA to RNA with GGAGA (Supplementary Fig. 8).

Our results suggest that RelA stabilizes an initial complex of RNA associated with Hfq monomer and thereby enables attachment of additional monomers to form Hfq hexamers. To show that preliminary RNA binding to Hfq monomers is necessary and sufficient for RelA to initiate Hfq assembly, we mixed limiting levels of wild type Hfq with comparatively higher levels of either Hfq distal mutants (I30D or G29A) and *sodA*

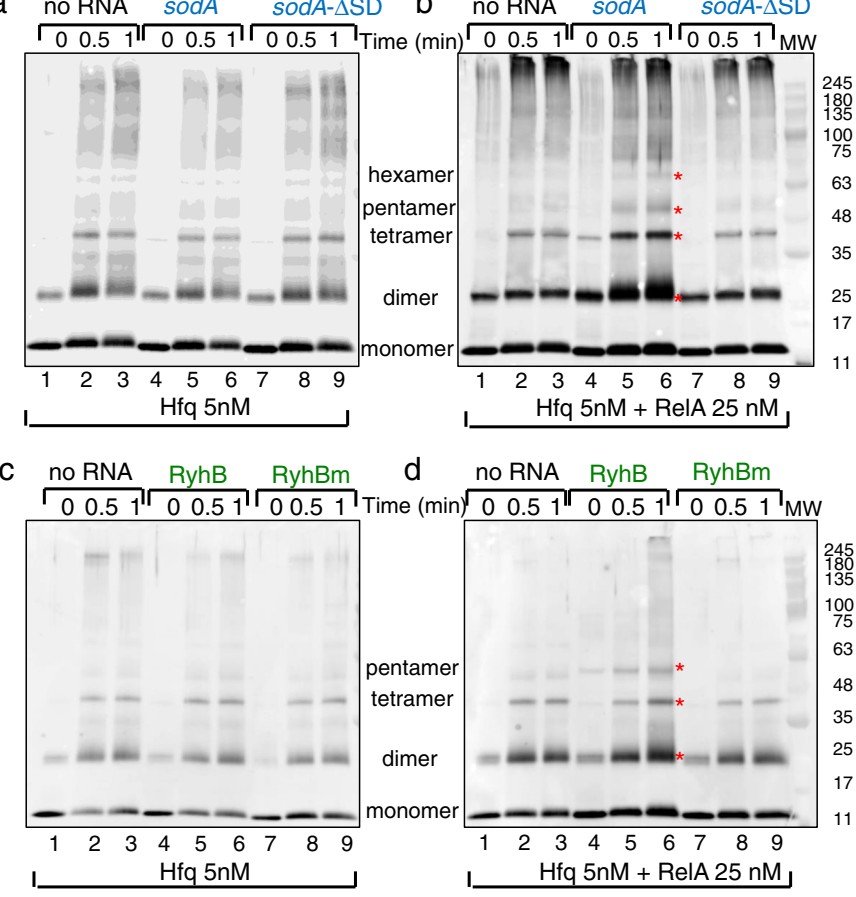

**Fig. 4 RelA-mediated Hfq assembly requires interaction with GGAGA sequence (western using α Hfq antibody). a, c** In the absence of RelA, RNA has no effect on Hfq multimerization. Reactions of Hfq incubated without or with RNA (100 nM) at 22 °C for 10 min were UV crosslinked followed by protein crosslinking with 0.2% of glutaraldehyde (Time in min indicates duration of protein crosslinking). The proteins separated in 4–20% MOPS gradient gels were detected using α Hfq antibody. **b, d** RelA induces Hfq multimerization when presented with RNA carrying GGAGA. Reactions of Hfq incubated with RelA, without or with RNA including RyhB (90 nt), RyhBm (90 nt) *sodA* (98 nt), and *sodA*-ΔSD (47 nt) were treated as in **a**. Asterisk indicates the formation of new Hfq multimers detected using wild type RNAs in the presence of RelA. Source data provided as a source data file.

(distal RNA unable to bind Hfq distal mutant) or with Hfq proximal face mutant (D9A) and RyhB (proximal RNA unable to bind Hfq proximal mutant). We based this experiment on the assumption that RelA stabilization of the complex formed by the binding of RNA to wild type Hfq monomers allows additional mutated Hfq subunits to join the initial complex to form hexamers. Figure 5b, c shows that RelA fails to facilitate assembly of extremely low inactive levels (<5 nM) of wild type Hfq when incubated along with *sodA* or RyhB RNA (lanes 4–6), all the more so of Hfq:G29A (distal) mutant incubated with *sodA* (Fig. 5b; lanes 10–13) and Hfq:D9A (proximal) incubated with RyhB (Fig. 5c lanes 10–13). However, mixing labeled *sodA* RNA with low, inactive levels of wild type Hfq (0.5 nM), and 4.5 nM of distal face Hfq:G29A subunits that are unable to bind *sodA* resulted in *sodA* binding and heterogeneous complex formation (Fig. 5b; lanes 14), indicating that the little RNA binding of *sodA* by wild type Hfq monomers enabled stabilization of the complex by RelA. The binary complex served as an anchor for further attachment of Hfq:G29A subunits leading to the formation of a mixed subunits hexamer (see illustration in Fig. 5e). Likewise, mixing inactive levels of wild type Hfq with Hfq:D9A proximal face mutant that is unable to bind RyhB resulted in RyhB binding (Fig. 5c; lanes 14–17) further confirming that initial RNA binding to wild type Hfq monomer is necessary and sufficient to initiate hexamer formation by RelA.

The position of the complex formed by RNA binding to Hfq: I30D is slightly different from the one detected with wild type Hfq (Fig. 5d; lanes 3 and 9). Interestingly, we noticed that the position of the complex obtained by mixing wild type Hfq (0.5 nM) with Hfq:I30D (4.5 nM) is similar to that detected with Hfq:I30D alone (Fig. 5d; lanes 14, 15). However, as the number of Hfq wild type subunits increases the position of the complex is shifted toward the wild type position (Fig. 5d; lanes 16, 17 and illustration in Fig. 5f), further confirming that the hexamer is formed by mixing different subunits and as the ratio changes the complex's position changes too. Together the results strongly demonstrate that RelA stabilization of the preliminary complex of Hfq sub-unit bound by RNA enables the attachment of additional subunits to form hexamers.

The use of Hfq mutants further confirmed that RelA-mediated Hfq assembly requires an initial binding of RNA to Hfq monomers. RelA failed to induce the RNA binding activity of low levels (5 nM) of Hfq proximal mutants presented with proximal face RyhB sRNA (Supplementary Fig. 9a; lanes 7, 10) and Hfq distal mutants presented with distal face *sodA* RNA (Supplementary Fig. 9b; lanes 13, 16). In contrast, RelA induced the RNA binding activity of Hfq mutants presented with RNAs capable of binding the opposite face of the mutation (Supplementary Fig. 9a, b; lanes 13 and 7, 10). Interestingly, only low concentrations of RelA (25 nM) induced the RNA binding

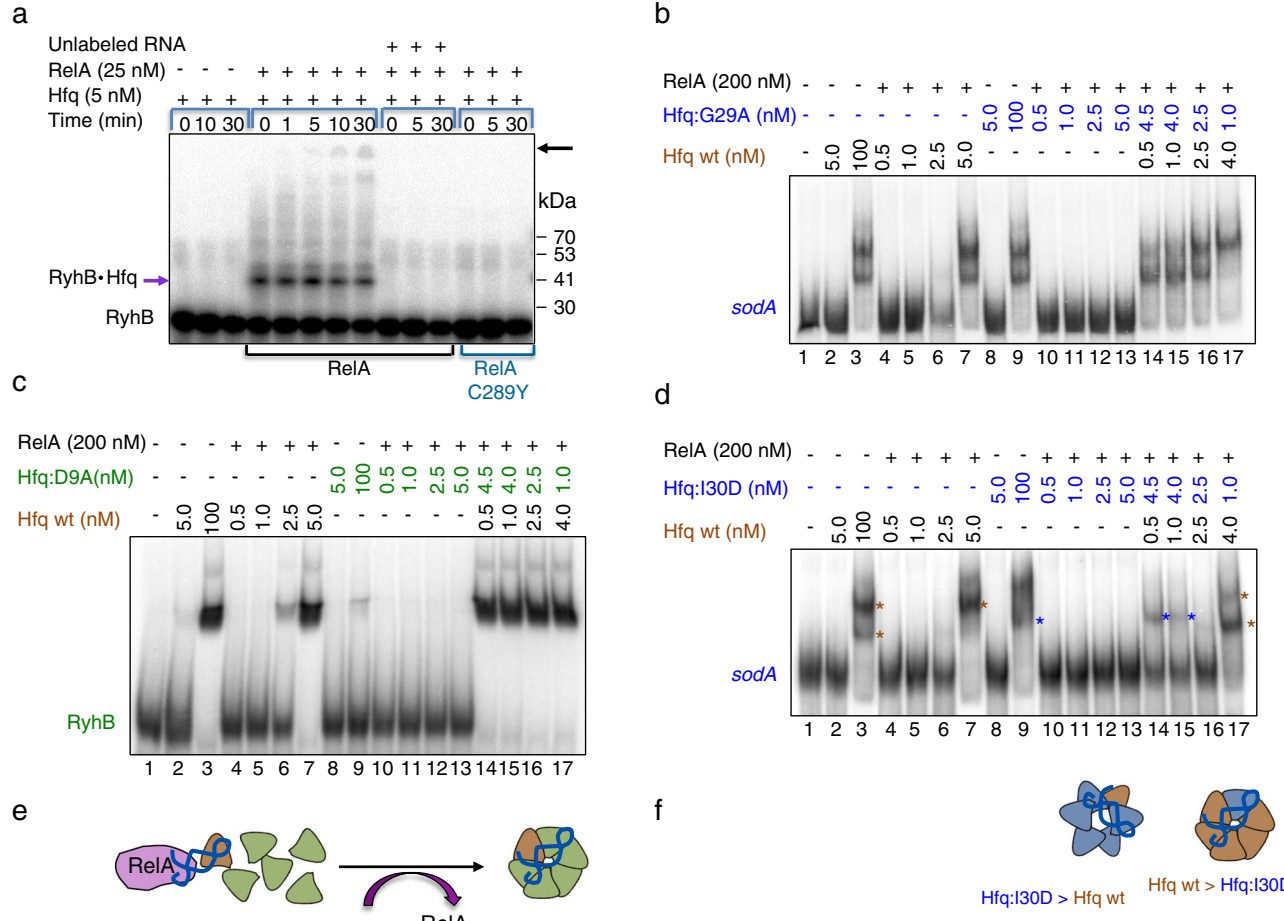

**Fig. 5 RelA stabilizes the binding of RNA to Hfq monomer and enables further assembly. a** RelA facilitates RyhB binding to Hfq monomers. Reaction mixtures of Hfq incubated for 10 min at 22 °C with labeled RyhB (1 nM; 90 nt) without or with RelA or RelA:C289Y were UV crosslinked (zero time point) followed by protein crosslinking with 0.2% glutaraldehyde. The crosslinking was stopped with 200 mM of fresh glycine and the products analyzed in 4–20% MOPS gradient gel. Unlabeled competitor RNA (100 nM) was added where indicated. The estimated MW of the RNA•Hfq monomer complex (purple arrow on lower left side) is ~40 kDa. Note that the addition of RNA alone to low levels of Hfq (5 nM) does not result in Hfq-RNA stable binding. Also, RelA:C289Y does not enable the binding of RNA to Hfq monomers. In the presence of RelA, as time of crosslinking progressed, higher forms of Hfq•RyhB emerged indicated by a black arrow. **b** Gel mobility shift assay carried out with different ratios of Hfq to Hfq:G29A distal mutant (blue) incubated with labeled *sodA* distal RNA (1 nM; 210 nt) for 10 min at 22 °C followed by 4% native gel electrophoresis. The binding of as low as 0.5 nM of Hfq to *sodA* RNA is necessary and sufficient to enable further assembly with Hfq:G29A subunits. See illustration of Hfq assembly pathway in **e**. **c** As in **b** except that RyhB (1 nM; 50 nt) and a proximal face Hfq mutant Hfq:D9A were used. See illustration of Hfq assembly pathway in **e**. **d** As in **b**, except that RNA binding to Hfq:I30D forms a complex that is different from that formed by wild type (see illustration of the two forms of hexamers in **f**. Brown asterisk denotes the position of the wild type complex (lane 3) whereas blue asterisk denotes the position of the Hfq:I30D complex (lane 9). The position of the complex in lanes 14 and 15 is shifted toward wild type as the ratio of wild type to I30D is increasing (lane 17). **e** Illustration of Hfq assembly induced by RelA. RelA stabilizes the binding of RNA to wild type Hfq monomer (brown) and enables the addition of mutant Hfq:G29A or Hfq:D9A subunits (green). **f** Illustration of Hfq hexamers composed mainly by wild type (brown) or by mutated Hfq:I30D subunits (blue). Source data provided as a source data file.

activity of Hfq:G29A distal mutant presented with RyhB (Supplementary Fig. 9c). Since the binding affinity of G29A to RyhB is significantly low as compared to Hfq wild type (Supplementary Fig. 9c; lanes 3, 15) we suspect that high RelA levels (200 nM) competed with Hfq:G29A for RyhB (Supplementary Fig. 9c; lanes 17, 18). Unlike G29A, only a high concentration of RelA facilitated Hfq:K56A-*sodA* binding. As the affinity of RelA for RyhB is much more pronounced than RelA affinity for *sodA*, we suggest that high RelA levels can compete with Hfq:G29A for the RyhB RNA but fail to compete with Hfq:K56A for *sodA* (Supplementary Fig. 9d).

Similarly, RelA facilitated formation of multimers including dimers, tetramers, and pentamers of Hfq proximal mutants in the presence of distal RNA and distal mutants presented with proximal RNA (Supplementary Fig. 10). In the absence of RelA,

the oligomerization pattern of Hfq mutants presented with *sodA* or RyhB was similar, indicating the importance of initial Hfq RNA binding for RelA-mediated oligomerization of Hfq (Supplementary Figs. 11 and 12).

**RelA ppGpp synthesis and RNA binding are mutually exclusive functions**. Given the observation that RelA:C289Y mutant failed to enable repression regulation of RyhB targets by RyhB yet it produced (p)ppGpp, we investigated the interaction between these two functions (Fig. 1a and Supplementary Figs. 1, 3 and 13a). In vivo assays of (p)ppGpp production carried out with chromosomally encoded *relA*+ and *relA*:C289Y strains showed that upon amino acid starvation both strains produced similar levels of (p)ppGpp. In the presence of a plasmid expressing RyhB,

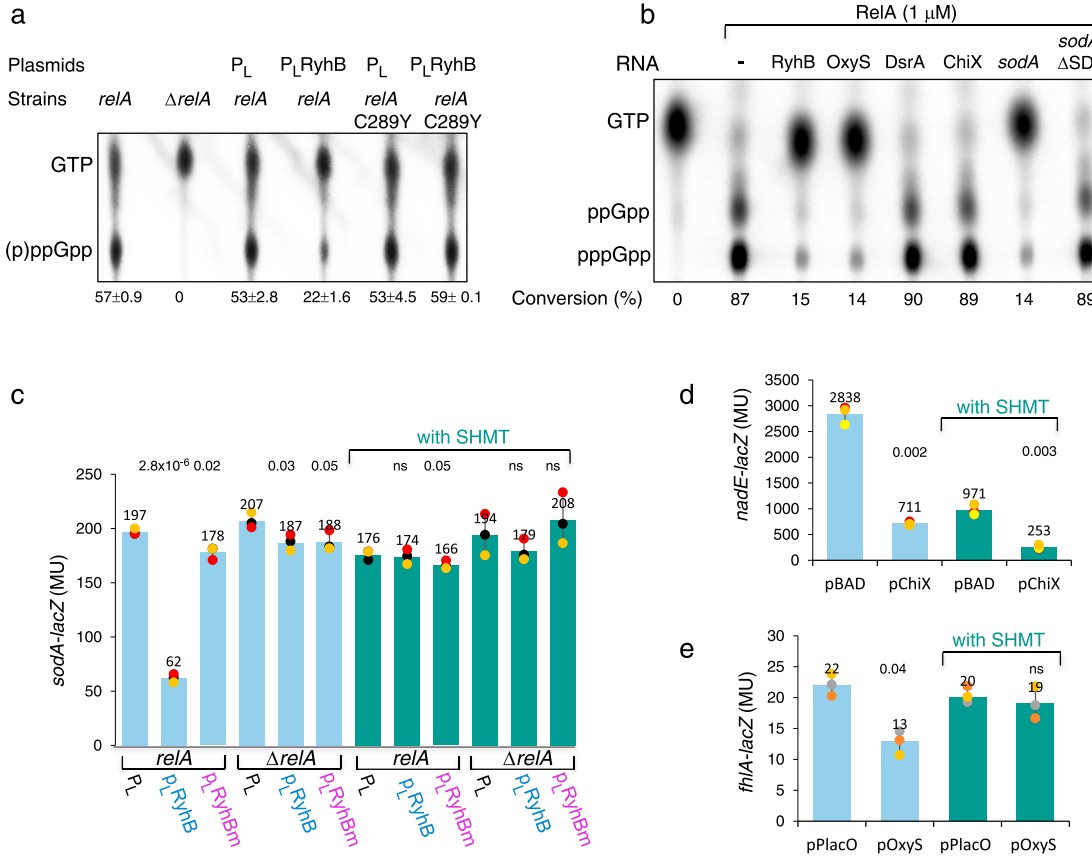

**Fig. 6 (p)ppGpp production and RNA binding are two mutually exclusive functions of the stringent response regulator RelA. a** In vivo (p)ppGpp production is inhibited by RyhB. *E. coli* strains; *relA*[+], Δ*relA*, and *rel*:C289Y (chromosomally encoded) carrying plasmids as indicated were assayed for (p)ppGpp production as described in "Material and methods". The intensity of the spots was determined by the ImageLab program and percentage of (p)ppGpp production of the total was calculated (% conversion). Mean and SD of two biological samples are presented. **b** In vitro (p)ppGpp production is inhibited by specific RNAs. Purified RelA was incubated with 50 nM of either RyhB, OxyS, DsrA, ChiX, *sodA*, or *sodA*-ΔSD and assayed by TLC. The intensity of the spots was determined by the ImageLab program and percentage of (p)ppGpp production of the total was calculated (% conversion). DsrA and ChiX lack a GGAGA site (see Fig. S5). **c–e** RelA mediated basepairing regulation under normal growth conditions and in response to amino acid starvation by serine hydroxamate (with SHMT). β-galactosidase assays of target gene fusions in the presence of their corresponding sRNAs (RyhB/*sodA*, ChiX/*nadE*, or OxyS/*fhlA*). ChiX expression was induced by 0.2% arabinose from the BAD promoter. Constitutive expression of plasmid encoded RyhB and OxyS in Δ*ryhB* and wild type, respectively. *n* = 3 biologically independent experiments. Data are presented as mean values ± SD. Two-tailed unpaired *t*-test were performed; the absolute *p* values are indicated. Source data provided as a source data file.

(p)ppGpp production by RelA decreased by about two-fold, whereas the production by RelA:C289Y was unaffected by the RyhB RNA, indicating that RNA binding inhibits the synthetase activity of RelA and further confirming that RelA:C289Y is unable to bind RNA (Fig. 6a).

Conversely, conditions of amino acid starvation induced by SHMT inhibited RelA regulatory activity of basepairing RNAs. β-galactosidase assays of *sodA-lacZ* carried out in the presence of SHMT showed that upon starvation, RelA-mediated repression regulation by RyhB was impaired (Fig. 6c).

The observation that RyhB affected the synthetase activity, prompted us to examine whether RelA is specific to RyhB and its targets. In vitro (p)ppGpp assay carried out with RelA incubated with OxyS sRNA carrying GGAG or with DsrA and ChiX sRNAs that lack this specific sequence (see sequence in Supplementary Fig. 5) showed that RNAs with GGAG or GGAGA (RyhB, OxyS, and *sodA*) decreased production of (p)ppGpp by RelA (Fig. 6b and Supplementary Fig. 13c). In contrast, DsrA, ChiX, and *sodA*-ΔSD had very little to no effect on (p)ppGpp production (Fig. 6b and Supplementary Fig. 13c). Furthermore, (p)ppGpp production in the presence of *sodA*m in which the GGAGA sequence was mutated to ACUCU was unaffected (Supplementary Fig. 13b).

Taken together the results demonstrate that RelA binds RNAs with GGAG and this binding interferes with its synthetic activity.

Given that RyhB and both its targets *sdhC* and *sodA* carry GGAG (Supplementary Fig. 5), RelA regulation of basepairing RNAs could be due to RelA binding of either RyhB or its targets or both. To examine which of the RNAs triggers RelA regulation we mutated the GGAG RelA binding site in RyhB. To make sure that the core sequence that is responsible for RyhB regulation of its targets remained intact, we examined expression of *sodB* whose expression is RelA-independent[27]. RNA analysis showed that both RyhB and RyhBm repressed *sodB* expression indicating that mutating GGAG had no effect on the core domain of RyhB (Supplementary Fig. 14a). Yet, RelA-mediated repression regulation of *sodA* by RyhBm was null (Fig. 6c). As the target mRNA *sodA* harbors an intact RelA binding site, RelA regulation of basepairing RNAs depends on RelA interacting with GGAG carried by sRNAs rather than by mRNA.

Moreover, β-galactosidase assay of the ChiX/*nadE* pair of which both lack the site GGAG (Supplementary Fig. 5) showed that induction of RelA synthetic activity had no effect on *nadE* repression regulation by ChiX (Fig. 6d). In the regulatory pair OxyS and *fhlA* of *E. coli*, only OxyS carries GGAG

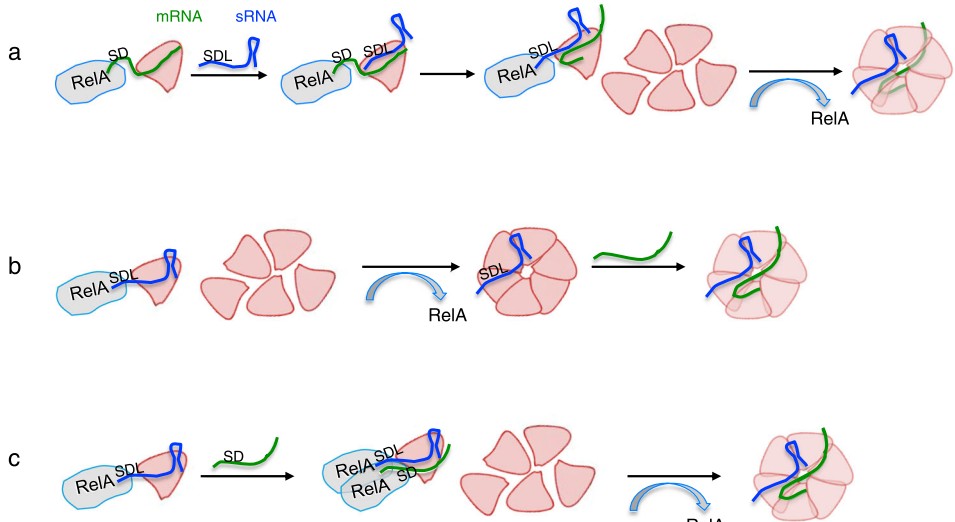

**Fig. 7 RelA-mediated hexamerization of Hfq requires an initial binding of RNA to Hfq monomer.** sRNA (blue); mRNA (green); Hfq subunits (pink). **a** RelA binds and stabilizes an unstable complex of mRNA-Hfq monomer leading to subsequent sRNA binding while RelA switches from SD of the mRNA to the GGAG site (SDL Shine–Dalgarno Like) in the sRNA promoting further stabilization of the complex and enabling Hfq hexamerization. **b** RelA stabilizes an unstable complex of sRNA-Hfq monomer by binding the GGAG (SDL) site in the sRNA leading to Hfq hexamerization followed by target mRNA binding. **c** RelA stabilizes an unstable complex of sRNA-Hfq monomer by binding the GGAG (SDL) site in the sRNA. The binding of SD site in the mRNA possibly by another RelA promotes further stabilization and ultimately Hfq hexamerization. In all cases, sRNA mediated mRNA regulation is expedited by RelA binding of sRNAs.

(Supplementary Fig. 5). β-galactosidase assay of the OxyS/*fhlA* showed that upon induction of RelA-synthetic activity, OxyS no longer repressed expression of *fhlA-lacZ* indicating that GGAG site of OxyS is sufficient to enable regulation (Fig. 6e). Thus, sRNAs with GGAG facilitate the base-paring function of RelA in vivo.

## Discussion
**The pathway by which RelA mediates Hfq assembly.** In this study we show that Hfq monomer binds RNA in the presence of RelA, challenging the previous concept that Hfq binds RNA only as a hexamer. These results support the notion that RNA binding coincides with Hfq hexamerization which requires initial disassembly of Hfq followed by RNA binding to individual Hfq subunits to form a new RNA-bound complex.

By mixing limiting amounts of wild type and high levels of Hfq mutant with RNA that can be bound only by wild type, we discovered that preliminary RNA binding to Hfq monomers is necessary and sufficient for RelA to initiate Hfq assembly, thereby forming mixed oligomeric sub-unit complexes. Together, these results led us to propose that RelA by stabilizing the originally unstable complex of RNA bound to an Hfq monomer enabled the attachment of additional subunits to form hexamers (model Fig. 7). Having said that our previous results showing that RelA of 70S extracts enables binding of labeled RNA by Hfq monomer indicate that the ribosomes provide a supporting environment for RelA-RNA-Hfq monomer binding, possibly by creating a micro-environment that holds the components in the correct position and structure[27].

Our in vitro experiments show that the affinity of RelA to sRNA is higher than its affinity for mRNA carrying the same GGAG sequence. Our in vivo *lacZ* assays demonstrate that RelA binding to sRNAs is essential for target gene regulation. The preference of RelA observed in vitro for sRNAs with GGAG to mRNAs with the same sequence is not clear. As the GGAG sequence resembles the SD sequence present in many mRNA targets, it is plausible that in vivo, RelA binds the SD sequence of mRNAs (Fig. 7a). In this scenario, RelA binds and stabilizes an unstable complex of mRNA-Hfq monomer leading to subsequent sRNA binding while RelA switches from SD of the mRNA to the GGAG site in the sRNA promoting further stabilization of the complex and enabling Hfq hexamerization. Alternatively, RelA stabilizes an unstable complex of sRNA-Hfq monomer by binding the GGAG site in the sRNA leading to Hfq hexamerization followed by target mRNA binding (Fig. 7b). It is also possible that the complex of RelA-sRNA-Hfq is further stabilized via binding of the target mRNA SD site by a second RelA leading to Hfq hexamerization (Fig. 7c). In either case, sRNA mediated mRNA regulation is expedited by RelA binding of sRNAs.

The proposed mechanism assumes that RelA is effective in sub-stoichiometric amounts relative to Hfq, which in turn corresponds to the previously reported low intracellular concentration of RelA[38]. By quantifying absolute protein synthesis rates under three different growth conditions in exponential phase, Li et al.[38] estimated RelA concentration to be in the range of 200–330 nM. Our assessment of the concentrations of chromosomally and plasmid encoded RelA in exponential phase is $518 \pm 150$ nM and $2 \pm 1$ μM, respectively, whereas Hfq monomer concentration carried out under the same conditions show that *relA+* and *relA−* strains harbor ~4.5 and 2.5 μM, respectively (Supplementary Fig. 15). We also measured the stationary phase intracellular concentrations of RelA ($2.2 \pm 0.15$ μM) and Hfq ($8.5 \pm 3$ μM) (Supplementary Fig. 16b), which shows that in between these growth phases RelA increased by four-fold, whereas Hfq increased by two-fold. The significant increase in RelA expression in stationary phase is due to the stationary phase dependent P2 promoter of RelA[39,40]. These measurements indicate that the levels of Hfq are nine and four-fold higher than the levels of RelA in exponential and stationary phase, respectively. Yet, the absolute concentration of Hfq is not indicative of Hfq availability. Co-IP studies have revealed thousands of Hfq-bound RNAs and overexpression of Hfq-dependent sRNAs resulted in the sequestration of Hfq and thus in Hfq depletion[13,41–43]. Here, we show that RelA enables binding of RNAs by otherwise ineffective amounts of Hfq in vitro, and facilitates Hfq-mediated basepairing regulation of specific sRNA/

mRNA pairs in vivo, indicating that under specific conditions and/or environments, Hfq availability is inadequate.

Based on our Co-IP studies we estimate that about 50% (±5) of Hfq can interact with RelA in cell lysates (Supplementary Fig. 16a). Given that the number of molecules of RelA and Hfq per cell before Co-IP were estimated to be 4000 and 15,360, respectively, the ratio of the Hfq fraction (7680 molecules) interacting with RelA (4000 molecules) is around 2, suggesting that one molecule of RelA interacts with two molecules (monomers) of Hfq (Supplementary Fig. 16b).

**RelA N-terminal domain binds RNA.** Deletion and mutational analyses revealed that RelA N-terminal domain is responsible for mediating Hfq-sRNA based target gene regulation. Specifically, two single-point mutations in RelA (C289Y and T298I) that clustered in one helix of the RelA N-terminal domain failed to enable repression of RyhB target gene fusions. Unlike wild type RelA that facilitated RNA binding of low concentrations of Hfq by triggering Hfq oligomerization, RelA C289Y and T298I mutants showed no effect on Hfq RNA binding nor they affected Hfq oligomerization. As these RelA mutants produce (p)ppGpp similar to wild type, (p)ppGpp production, and RelA-mediated base-pairing RNA regulation, although both reside in the N-terminal domain were found to be distinct functions. Previously, using a highly sensitive binding assay, we found that incubation of His-tagged RelA-CTD purified from wild type *hfq* cells (a gift from G. Glaser) with RyhB resulted in residual binding of Hfq to RyhB. As RelA-CTD was co-purified with Hfq, we suggested that this portion of the protein might also act as a functional partner of Hfq[27]. Our current in vivo and in vitro genetic and biochemical studies demonstrated the importance of the RelA N-terminal domain for Hfq assembly as opposed to its C-terminal domain. Therefore, we suspect that RelA-CTD purified from wild type Hfq cells was contaminated with Hfq due to the presence of 24 histidine residues at its C-termini[44]. Unlike the intact RelA protein, which was further investigated by its purification from Δ*hfq* cells, we did not explore the function of this domain any further.

**RelA sRNA binding and (p)ppGpp production are mutually exclusive functions.** An interaction between RelA-like protein and RNA was documented previously for RelQ[45]. The authors showed that the small alarmone synthetase RelQ from the Gram-positive pathogen *Enterococcus faecalis* bound mRNAs at AGGAGG sites. The enzymatic activity of *E. faecalis* RelQ was inhibited by mRNA binding, and addition of (p)ppGpp counteracted the inhibition. Because (p)ppGpp synthesis and (p)ppGpp binding were mutually incompatible with RelQ:RNA complex formation, it was proposed that RelQ enzymatic and RNA binding activities are subject to allosteric regulation. Likewise, we propose that RelA (p)ppGpp synthesis and RNA binding are two mutually exclusive functions due to allosteric inhibition.

**RelA binds RNAs with GGAG.** Comparing RyhB and RyhBm levels in *relA*$^+$ and *relA*$^-$ strains indicate that RelA binding results in sRNA stabilization (Supplementary Fig. 14b). The level of the wild type RyhB RNA was higher in *relA*$^+$ than in *relA*$^-$, whereas the levels of RyhBm in which RelA GGAGA binding site was changed were similar in both *relA*$^+$ and *relA*$^-$. Moreover, qRT-PCR analysis revealed that the levels of RyhB were ~2-fold higher in *relA*$^+$ than in RelA mutants impaired in RyhB target regulation, indicating that RelA-sRNA binding allows moderate stabilization of the associated RNAs (Supplementary Fig. 14c).

Co-immunoprecipitation using α RelA antibody to detect whether Hfq was precipitated in a complex with RelA further confirmed that in vivo, RNA links between RelA and Hfq forming a complex. As Hfq did not precipitate with RelA:C289Y mutant, it strongly supported the notion that RelA binds RNA that is bound by Hfq. Further RelA-RNA binding assays demonstrated that RelA binds RyhB with a higher affinity compared to *sodA* and footprinting revealed that RelA binds and therefore protects a specific sequence of GGAG.

Quantitative RT-PCR analysis of the RNA bound to RelA after Co-IP revealed that sRNAs carrying GGAG sequence including RyhB, SraC, and McaS were bound by RelA but not by RelA: C289Y (Supplementary Fig. 17). The calculated copy number of these sRNAs was significantly higher in lysates of wild type *relA* than in lysate of *relA*:C289Y. The copy number of MgrR and MicC sRNAs that lack the GGAG sequence was comparable in both *relA* and *relA*:C289Y, whereas *sdhC* and *sodA* mRNA levels carrying GGAG were somewhat lower in *relA*:C289Y compared to wild type RelA, indicating that GGAG targets may play a role in RelA-mediated RNA regulation. Before Co-IP, both GGAG sRNAs and those lacking the site displayed similar levels, whereas *sodA* and *sdhC* RyhB targets levels were lower in wild type RelA cells compared to *relA*:C289Y mutant, due to RelA-mediated repression by RyhB (Supplementary Fig. 17).

**The physiological conditions leading to RelA regulation of basepairing RNAs.** Our data indicate that RelA is specific to sRNAs with GGAG and that it affects not all sRNA/mRNA pairs. However, what distinguishes the groups is unclear. Conceivably, the affinity of Hfq for the RelA-independent class of sRNAs and/or mRNAs is high and therefore also low levels of Hfq are effective. For example, the binding affinity of the GGAG sequence lacking sRNAs such as ChiX, DsrA, RprA, MgrR, MicA, MicC, MicF, and SpoT42 was estimated by filter binding or gel mobility shift assays to be 0.21, 0.54–23, 25, 0.48, 2.3, 3.3, 1.7, and 20 nM, respectively[17,24,46–48]. Also, ChiX, MgrR, and DsrA that lack the GGAG sequence were reported to be better competitors for binding Hfq than RyhB, McaS, or CyaR sRNAs[47]. Interestingly, both MgrR and ChiX carry in addition to poly(U) tail three and four ARN motifs, respectively[25,46]. Deleting these motifs led to a significant loss of stability of both sRNAs, while adding these motifs to RyhB increased its stability[25]. Thus, indicating that additional binding of the sRNAs to Hfq distal site by the ARN motifs apart from the proximal site interaction results in enhanced affinity and hence stability of the sRNAs.

In investigating the occurrence of sRNAs with GGAG, we identified 26 sRNAs with this sequence from a group of 86 (Supplementary Fig. 18). Using a combination of the Clustal omega (https://www.ebi.ac.uk/Tools/msa/clustalo/) and Genedoc software, we identified 19 sRNAs in which the GGAG site was conserved in several bacterial species (Supplementary Fig. 19). Of the 26 sRNAs, many are expressed during stationary phase and/or in minimal media such as McaS, RybB, RydB, RyfD, RyeA/SraC, RyhB, and GadY. A few are generated from the 3′UTR of protein coding genes (*glnA*, *kilR*, *malG*, *allR*). Intriguingly, a significant number of the sRNAs belongs to type I toxin-antitoxin (TA) systems (RalA, SibA, SibC, SibD, SibE, SokC, and SokE). Among these TA systems, only RalA is known to be stabilized by Hfq[49]. It may be that the Hfq binding affinity of these sRNAs is extremely low and almost undetectable, thus requiring RelA assistance. Alternatively, RelA binding of these sRNAs plays a regulatory role in the absence of Hfq. As RelA synthetase activity is incompatible with RelA regulation of base-paring RNAs, it is intriguing to speculate that upon normal growth conditions, RelA facilitates repression of the TA systems to decrease toxicity, whereas under conditions of amino acid starvation, RelA indirectly leads to an increase in expression of the toxin genes of the TA systems thereby modulating primary metabolic pathways.

## Methods

**Bacterial strains and plasmids.** Strains, plasmids, and primers used in this study are listed in Tables S1–S3. Bacteria were grown routinely at 37 °C in Luria-Bertani (LB) medium. Ampicillin (amp, 100 μg/ml), kanamycin (kan, 40 μg/ml), chloramphenicol (cm, 30 μg/ml), and tetracycline (tet, 10 μg/ml) were added where appropriate.

**Strain construction.** Chromosomal gene deletion mutants were carried out using the kanamycin and chloramphenicol cassettes of pKD4 and pKD3, respectively[50,51]. The chromosomal deletions were transferred into fresh genetic background by transduction using the P1 bacteriophage. To construct Δ*relA::kan*, primers 2187 and 2188 were used to replace 4 kb region encompassing *relA1* with 1.2 kb of kanamycin cassette. Δ*ryhB::cam* was constructed using primers 618 and 619. Δ*hfq::cam* was constructed using primers 2383 and 2384. To generate Δ*relA::frt*, Δ*hfq::frt*, and Δ*ryhB::frt*, the antibiotic resistance genes were removed using pCP20[50].

**Plasmid construction.** The *relA* gene was amplified from *E. coli* MC4100 chromosomal DNA using the primers 2530 and 2345. The PCR product was digested with the *Pst*I and *Hind*III restriction enzymes and ligated downstream of P_BAD_ of p15A vector generating pRelA. To construct pRelA-ΔCTD (2971-2972) and pRelA-ΔNTD (2973-2974), whole plasmid PCR was carried out using primers as above and pRelA as template. Point mutations in the *relA* gene including C289A (3039-3040) and Q264E (3134-3135) were introduced in pRelA plasmids using the Gibson cloning method[52]. The pRelA:C289Y;Y290C plasmid was generated by introducing the Y290C mutation (3036-3037) in the pRelA:C289Y plasmid (obtained by random mutagenesis). For purification of the RelA protein, we cloned the *relA* gene in pET-15b vector by the Gibson cloning method. To this end primers 3031-3032 were used to amplify pET-15b vector and primers 2999-3000 were used to amplify the *relA* gene. RelA point mutations; C289Y (3069-3070), C289Y;Y290C (3036-3037), and C289A (3039-3040) were cloned in pET-15b using the same method (Gibson). For Hfq protein purification, the *hfq* gene was amplified using the primer pairs 2687-2351 and cloned in the pET-15b vector using *Nco*I/*Bam*HI restriction enzymes (PT7-Hfq). Hfq point mutations; D9A (3168-3169), K56A (3048-3049), G29A (3170-3171), and I30D (3054-3055) were introduced in PT7-Hfq using Gibson cloning method. P_L_-RyhBm was constructed by whole plasmid PCR using primers 3266-3267 and P_L_-RyhB as template. The *chiX* gene fragment of SL1344 was amplified using primers 2594 and 2595 and sub-cloned downstream of P_BAD_ into the unique *Eco*RI and *Hind*III sites of pJO244. To construct *nadE-lacZ* translational fusion in pSC101 (pBOG552), *nadE* 5′-end fragment carrying 166 nucleotides from −327 upstream of the AUG initiation codon to +164 was amplified using primers 2907-2908 and subcloned into the unique *Eco*RI and *Bam*HI sites of pBOG552. To construct *sodA-lacZ* translational fusion, the *sodA* 5′-end fragment carrying 391 nucleotides from −293 upstream of the AUG initiation codon to +98 was amplified from MC4100 chromosomal DNA by PCR using oligonucleotides 2927 and 2928 and subcloned into the unique *Eco*RI and *Bam*HI sites of pRS552[53]. To construct MC4100 *sdhC-lacZ* translational fusion, the *sdhC* 5′ end fragment carrying 319 nucleotides from −281 upstream of the AUG initiation codon to +38 was amplified from MC4100 chromosomal DNA by PCR using oligonucleotides 2491 and 2505 and subcloned into the unique *Eco*RI and *Bam*HI sites of pRS552. The *lacZ* fusions were then recombined onto λRS552 and integrated into the attachment site of *relA*⁺Δ*ryhB::frt* (A-506), Δ*relA::frt* (A-1036), and Δ*relA::frt* Δ*ryhB::frt* (A-1046).

**Random mutagenesis.** Random mutagenesis of the *relA* gene (pRelA; p15A) was carried out using hydroxylamine as described before[54]. The mutagenized plasmids were transformed into MC4100 Δ*relA* strains carrying *sdhC-lacZ* chromosomal fusion and a plasmid expressing RyhB (P_L_-RyhB; ColE1). Blue versus white colonies were selected on LB plates containing 40 μg/ml of 5-bromo-4-chloro-3-indolyl-β-D-galactopyranoside (X-Gal) and 0.1% arabinose.

**Scarless point mutations in the chromosome.** Chromosomal scarless point mutations within *relA* were carried out as described in[55]. Briefly, the *tetA-sacB* cassette from the XTL634 strain chromosome was amplified using the primer pairs 3112-3113 carrying sequences homologous to the *relA* gene. The PCR product was inserted into *relA*⁺*hfq*⁺(A-397) and *relA*⁺Δ*hfq::frt* (A-950) generating the *relA::tetA-sacB* strain. Next, PCR product generated using primer pairs 3114-3115 that amplify the C289Y mutation from the (p_BAD_-RelA; p15A) was used to transform *relA::tetA-sacB*. Colonies sensitive to tetracycline were selected on fusaric acid containing plates.

**β-galactosidase assays.** Strains as indicated were grown for 16–18 h in M9 minimal medium containing 0.4% glycerol, 0.04% glucose, and 0.2% arabinose for induction of RelA expression. The cultures reached OD600 of ~0.3–0.4. LacZ assays were carried out as previously described[54]. To determine the effect of RelA-mediated regulation of RyhB, ChiX, and OxyS target genes under amino acid starvation, the strains as indicated were grown for 16–18 h in MOPS minimal medium supplemented with 0.4% glycerol and 0.04% glucose and 0.2% arabinose (for ChiX induction) in either the presence or absence of 500 μg/ml of Serine hydroxamate (SHMT). LacZ assays were carried out as above.

**RelA protein purification.** BL21-DE3 Δ*relA* Δ*hfq* strains carrying pET-15b plasmids expressing the 6X His-tagged RelA wild type, RelA:C289Y, RelA:C289Y;Y290C and RelA:C289A grown in 1lt of LB at 37 °C to OD600 of 0.5–0.6 were treated with 1 mM IPTG and continued to grow to OD600 of 2. The pellets were washed once in 1X PBS and then dissolved in 30 ml of cold buffer A {10 mM imidazole, 500 mM NaCl, 200 mM NaH₂PO₄ (pH 7.4)} along with 50 μl each of PMSF, DNase I (0.1 mg/ml) and 1 M MgCl₂. Following vortex, homogenization, and lysis in micro-fluidizer, the supernatant was collected by centrifugation and loaded onto His Trap Chelating HP column (1 ml) in the Ni-AKTA Prime machine. The column was washed with washing buffer {400 mM imidazole, 500 mM NaCl, 20 mM NaH₂PO₄ (pH 7.4)} and fractions of 8 ml were collected in the fraction collector. Fractions showing the maximum amounts of RelA protein were collected in a snakeskin dialysis bag and subjected to cleavage by TEV protease (overnight, 4 °C, dialysis in buffer A). The cleaved protein was passed through the column again and eluted with washing buffer. All Fractions were collected and subjected to dialysis overnight in RelA buffer {50 mM Tris-Ac (pH 8.5), 10 mM potassium phosphate buffer (pH 8.5), 10 mM EDTA, 1 mM DTT, 25% glycerol}.

**Hfq protein purification.** BL21-DE3 Δ*hfq* carrying pET-15b plasmids expressing Hfq wild type, Hfq K56A, Hfq I30D, Hfq D9A, and Hfq G29A grown in 1lt of LB at 37 °C to OD600 of 0.5–0.6 were treated with 1 mM IPTG and continued to grow to OD600 of 2. The pellets were washed once in 1X PBS and then dissolved in 50 ml of lysis buffer {50 mM Tris (pH 8), 1.5 M NaCl, 250 mM MgCl₂, 1 mM β-mercaptoethanol} along with 50 μl each of PMSF, DNase I (0.1 mg/ml) and 1 M MgCl₂. Following vortex, homogenization, and lysis in micro-fluidizer, the supernatant was collected by centrifugation, heated at 85 °C for 45 min, clarified by centrifugation, and treated with 30 μg/ml of RNase A for 1 h at 37 °C. After RNase A treatment, the supernatant was loaded onto His Trap Chelating HP column (1 ml) in the Ni-AKTA Prime machine. The column was washed with washing buffer {50 mM Tris (pH 8), 1.5 M NaCl (pH 7.4), 0.5 mM β-mercaptoethanol} and 8 ml fractions were collected in the fraction collector. Fractions showing the maximum amounts of Hfq protein were collected in a snakeskin dialysis bag and subjected to overnight dialysis in Hfq buffer {50 mM Tris (pH 7.5), 50 mM NH₄Cl, 1 mM EDTA, 10% glycerol}.

**Primer extension.** Total RNA (15 μg) extracted using Tri reagent (Sigma) from strains as indicated was incubated with end labeled *sodB* specific primer (810) at 70 °C for 5 min, followed by 10 min in ice. The reactions were subjected to primer extension at 42 °C for 45 min using 1 unit of MMLV-RT (Promega) and 0.5 mM of dNTPs. Extension products were analyzed on 6% acrylamide 8 M urea-sequencing gels.

**Northern analysis.** RNA samples (15 μg) isolated from strains as indicated were denatured for 10 min at 70 °C in 98% formamide loading buffer, separated on 6% acrylamide 8 M urea gels and transferred to Zeta Probe GT membranes (Bio-Rad laboratories) by electroblotting. To detect RyhB, the membrane was hybridized with end labeled RyhB primer (470) in modified CHURCH buffer (1 mM EDTA, pH 8.0, 0.5 M NaHPO4, pH 7.2, and 5% SDS) for 2 h at 45 °C and washed as previously described[56].

**In vitro RNA synthesis.** DNA templates for RNA synthesis: RyhB, 90 nucleotides used for crosslinking and footprint were amplified using primers 678-567; RyhB 50 nucleotides used for EMSA was generated using primers 678-3265; RyhB mutant, 90 nucleotides used for crosslinking was generated using primers 3272-567; *sodA* 210 and 56 nucleotides used for EMSA were generated using primers 1764-1765 and 3209-3211, respectively; *sodA* 98 nucleotides used for crosslinking and footprinting was generated using primers 1764-3203 and *sodA*-ΔSD 47 nucleotides used for crosslinking was generated using primes 1764-3198. The 98 nucleotides *sodAm* template for footprinting was obtained from TWIST Bioscience. RNAs were synthesized with phage T7 RNA Polymerase (25 units, NEB) in 50 μl reaction containing 1X T7 RNA Polymerase buffer, 10 mM DTT, 20 units of recombinant RNase inhibitor, 500 μM of each NTP, and 300 ng of T7 promoter containing template DNA at 37 °C for 2 h, followed by 10 min at 70 °C. Thereafter, Turbo DNase was added and the reactions were incubated at 37 °C for 30 min. The RNA was purified by phenol-chloroform extraction and then precipitated using 0.3 M ammonium acetate, ethanol, and quick precip. Labeled α-P³² ATP RNAs were generated using low concentrations of unlabeled ATP (20 μM).

**Electrophoretic mobility shift assay.** Reactions (10 μl) in binding buffer C {50 mM HEPES (pH 7.5), 10 mM MgCl₂, 100 mM NH₄Cl, and 1.5 mM DTT} carrying labeled RNA (1 nM) and/or Hfq and/or RelA as indicated were incubated at 22 °C for 10 min and analyzed on 4% native gels using native loading buffer.

**Protein crosslinking assay.** Purified proteins along with RNAs (100 nM) as indicated in the figures were incubated in binding buffer C {50 mM HEPES (pH 7.5),

10 mM MgCl$_2$, 100 mM NH$_4$Cl, and 1.5 mM DTT} at 22 °C for 10 min followed by UV crosslinking for 5 min (254 nm 20,000 μJ/cm$^2$). Samples were collected before and after the addition of 0.2% glutaraldehyde at the time points indicated in the figures. Reactions were stopped by adding 200 mM of fresh glycine followed by heating at 95 °C for 10 min in sample loading buffer. The proteins were separated in 4–20% MOPS gradient gel. Hfq was detected by Hfq specific antibody (western).

**UV crosslinking assay with RelA**. To determine the binding of RNA to RelA, purified RelA (wild type or C289Y mutant) proteins were incubated with 1 nM of labeled RyhB or *sodA* at 22 °C for 10 min, in binding buffer C followed by UV crosslinking for 5 min as above. Where indicated, 100 nM of unlabeled RNA as competitor RNA or RNAse A (100 μg/ml for 1 h at 37 °C) to remove unbound RNA were added. Proteins heated at 95 °C for 10 min in sample loading buffer were analyzed in 15% SDS-PAGE.

**DMS and RNase T1 footprinting**. DMS and RNase T1 footprinting reactions were carried out with slight modifications[57]. Briefly, 0.5 pmol of RNA was incubated with 5 pmol of RelA (wild type or C289Y mutant) at 22 °C for 10 min, followed by incubation with 0.1, 0.2, or 0.4U of RNase T1 (37 °C for 5 min) or 0.3% DMS (25 °C for 5 min) in their respective buffers. Reactions were stopped by phenol/ chloroform extraction in the presence of 5 μg of yeast t-RNA for RNase T1 and precipitated using 0.5 M NaCl and quick precip. Primer extensions to detect the products were carried out using 5′-end labeled RyhB (3273) and *sodA* (3203) primers. The products were separated in 6% acrylamide 8 M urea gels.

**Protein co-immunoprecipitation (Co-IP) assay**. Strains as indicated in the figure: *E. coli relA$^+$hfq$^+$*(A-397), *relA$^+$Δhfq::frt* (A-950), *relA::C289Y hfq$^+$*(D-1182) and *relA::C289Y Δhfq::frt* (D-1172) were grown overnight in M9 minimal medium containing 0.04% glucose and 0.4% glycerol. The pellets (50 OD total) were resuspended in 1 ml of lysis buffer (20 mM Tris pH 8.0, 150 mM KCl, 1 mM MgCl$_2$, 1 mM DTT). Centrifuged at 11,200 × g for 5 min in 4 °C. Thereafter the pellets were flash frozen in liquid nitrogen and thawed on ice. The cells were lysed using 800 μl of lysis buffer and 800 μl of glass beads (0.1 mm) in a bead beater with 30 s burst and intermittent chilling on ice for a total of 5 min. The lysates were collected by centrifugation while one half of the lysate was treated with 100 μg/ml of RNase A for 1 h at 37 °C. For Supplementary Fig. 16, one half of the lysate was used for "before Co-IP" analysis and the other half was used for carrying out the Co-IP with RelA antibody. Immunoprecipitation was carried out with 35 μl of rabbit anti-RelA antibody, incubated shaking for 1 h at 4 °C. Then, 75 μl of Protein A Sepharose beads (prewashed in lysis buffer) were added and the mixture was further incubated with shaking for 1 h at 4 °C. The beads were washed five times with lysis buffer, soaked in 1X SDS sample loading buffer, boiled for 5 min at 95 °C. The samples were analyzed in 4–20% MOPS gradient gel and the proteins detected using rabbit anti-Hfq.

**Isolation of RNA precipitated during Co-IP assay**. Co-IP was carried out as above except that the cells were resuspended in lysis buffer containing 20 units of RNase inhibitor. As indicated above, here also one half of the lysate was used to isolate RNA for "before Co-IP" analysis. The RNA was isolated from the sepharose beads by the TRI-reagent, followed by precipitation with isopropanol and Glycoblue. RNA pellets were further washed with ethanol, air dried, and dissolved in 15 μl of DEPC.

**Quantitative real-time PCR (qRT-PCR)**. RNA concentrations (obtained after Co-IP extraction) were checked by NanoDrop machine (NanoDrop Technologies). DNA contaminations in the RNA samples were removed by DNase treatment according to the instructions provided by the manufacturer (RQ1 RNase free DNase, Promega). cDNA was synthesized from 2 μg of DNA free RNA using MMLV reverse transcriptase and random primers (Promega). Quantification of the cDNA was carried out in the Rotor gene 3000A machine (Corbett) using the real-time PCR SYBR-green mix (Absolute SYBR GREEN ROX MIX, ABgene). Reactions and machine handling were carried out according to manufacturer's instructions. Genes tested for real-time PCR were RyhB (566-3273), MgrR (3305-3306), MicC (3307-3308), SraC (3309-3310), McaS (3313-3314), *sdhC* (3302-3303), and *sodA* (3301-1765). Primer designing was carried out according to the guidelines provided by the IDT PrimerQuest software (https://eu.idtdna.com/ PrimerQuest/Home/Index?Display=SequenceEntry). Secondary structure formation within each primer was determined by the IDT OligoAnalyzer software (http://eu.idtdna.com/analyzer/Applications/OligoAnalyzer/). A standard curve was obtained by carrying out PCR with serially diluted *E. coli* MC4100 genomic DNA. Copy number calculation of each gene present in each sample was analyzed by the Rotor gene analysis software 6.0.

**Western blotting**. To detect Hfq or RelA, protein samples were separated by either 15% acrylamide gel (RelA) or 4–20% MOPS gradient gel (Hfq). Thereafter, the proteins were transferred to nitrocellulose membrane (Genscript) by the Genscript eBLOT L1 fast wet protein transfer system, as suggested by the manufacturer. After transfer, the membrane was incubated at room temperature for 1 h (shaking) in blocking solution containing 4% BSA, 4% skim milk, and 1X TBST. The membrane

was rinsed once with 1X TBST followed by incubation at room temperature for 1 h (shaking) in 10 ml of 1X TBST containing 3% BSA and 20 μl (1:500 dilution) of rabbit anti-Hfq raised against a synthetic peptide (SSAQNTSAQQDSEETE) of Hfq CTD (HY-LABS) or rabbit anti-RelA antibody raised against the purified RelA protein ADAR BIOTECH. Following three times wash (10 min each) with 1X TBST, the membrane was incubated in HRP conjugated secondary antibody solution (1 μl in 10 ml of 1X TBST, 1:10,000 dilution, Abcam, ab6721) for 1 h with shaking. The membrane was then rinsed once with 1X TBST and the protein were detected by incubation in ECL solution (Advansta Western Bright) for 1 min followed by the use of Image Quant LAS 4000 mini software.

**In vitro (p)ppGpp assay**. In total, 1 μM of purified RelA protein (wild type or C289Y) preincubated with or without RNAs (50 nM) in binding buffer C at 22 °C for 10 min were further incubated at 30 °C for 1 h in 1X synthesis buffer {2.5 mM GTP, 20 mM ATP, 200 mM Tris-HCl (pH 7.4), 5 mM DTT, 50 mM MgCl$_2$, 50 mM NH$_4$Cl, 50 mM KCl, and 10 μCi α-P$^{32}$ GTP}. The products were analyzed by thin-layer chromatography (TLC) with 1.5 M of KH$_2$PO$_4$ (pH 3.4) running buffer. The intensity of GTP and (p)ppGpp spots were measured by the ImageLab software and % of (p)ppGpp production was calculated.

**In vivo (p)ppGpp assay**. The intracellular (p)ppGpp accumulation was determined according to the protocol described by[58]. Briefly, strains grown in LB to OD600 of 0.3 were washed in low phosphate (0.2 mM of K$_2$HPO$_4$) MOPS minimal medium supplemented with all amino acids except serine and inoculated (1:100) in the same medium and grown to OD600 of 0.2. The cells were labeled with 100 μCi/ml of {$^{32}$Pi} H$_3$PO$_4$, by incubation at 37 °C for 10 min, followed by 1 h induction of amino acid starvation by the addition of 500 μg/ml of SHMT. Thereafter, the cells were pelleted, washed with 10 mM Tris-HCl (pH 8.0) and resuspended in the same buffer (20 μl) followed by lysis using an equal volume of prechilled 13 M of formic acid with intermittent tapping for 15 min on ice. Cell debris were eliminated by spinning at 13,000 rpm for 6 min at 4 °C. The collected supernatants were spotted on TLC plates with 1.5 M of KH$_2$PO$_4$ (pH 3.4) running buffer. The intensity of GTP and (p)ppGpp spots were measured by the ImageLab software and % of (p)ppGpp production was calculated.

**Measurement of intracellular RelA and Hfq levels**. *relA$^+$hfq$^+$*, *ΔrelAhfq$^+$*, *relA$^+$Δhfq*, and *ΔrelA*/(pRelA) strains were grown in M9 minimal medium containing 0.04% glucose and 0.4% glycerol (0.2% arabinose was added to induce RelA expression). At OD600 of 0.4, CFU/ml was calculated. Cell lysates were obtained by boiling the culture pellets with 1X laemmli sample buffer. RelA and Hfq protein levels in whole cell lysates were detected using α-RelA or α-Hfq antibody in 15% or 4–20% SDS-PAGE gels, respectively.

**Statistics and reproducibility**. Experiments presenting bar graphs with standard deviation and *p* value measurements were carried out using 3–5 bacterial colonies (as stated in the legends). Figures with representative gels/blots and micrographs were repeated at least twice (usually 2–4 times).

**Reporting summary**. Further information on research design is available in the Nature Research Reporting Summary linked to this article.

## Data availability
The authors can confirm that all relevant data are included in the paper and/or its Supplementary information files. The list of figures that have associated raw data: Figs. 1–6 and Supplementary Figs. 4, 7–11, and 13–16 as well as an Excel file with the calculations for all the Bar diagrams. The list of sRNAs provided in Supplementary Fig. 18 was generated using Biocyc.org with their b-number and GO-number as provided. Source data are provided with this paper.

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

## Acknowledgements

We are grateful to Sarah Woodson for her advice throughout the study. We appreciate the excellent assistance in strain and plasmid construction by Tal Hershko-Shalev and Noa Nur. This work was supported by: the Israel Science Foundation founded by The Israel Academy of Sciences and Humanities (138/18), and the Deutsch-lsraelische Projektkooperation (AM 441/1-1 SO 568/1-1).

## Author contributions

P.B. performed the experiments and helped wring the manuscript; F.H., M.K., and R.W. helped purifying the proteins; M.E.-W. helped in conceptualization of the experiments and reviewing of data. S.A. was responsible for conceptualization of the experiments, reviewing of data, writing the manuscript and funding.

## Competing interests

The authors declare no competing interests.
