## [Peer Review File · Nature Communications]

REVIEWER COMMENTS

Reviewer #1 (Remarks to the Author):

In their presented manuscript, Basu et al report that the (p)ppGpp synthase and stringent response regulator RelA can bind to Shine-Dalgarno-like sequences (GGAG) in certain small RNAs and mRNAs of *E. coli* and thereby facilitates hexamerization of the RNA chaperone Hfq and Hfq-mediated sRNA-mRNA regulation. This is a very well written manuscript and well-designed study that provides novel insights into the underlying molecular mechanisms of Hfq and importantly also additional factors in sRNA-mediated regulation in bacteria. While Hfq is well known as an RNA binding protein that facilitates sRNA-mRNA interactions during post-transcriptional regulation, there are still many open questions about the underlying molecular basis of assembly or competition of RNAs on Hfq and very little is known if additional partners are involved in mediating or regulating sRNA-mRNA interactions on Hfq. Typically, Hfq mediated post-transcriptional regulation is assumed to occur via sRNAs/mRNA duplexes bound to an Hfq homohexamer. Here, the authors address the important and still open question in the field, whether RNAs can only bind to a pre-assembled Hfq hexamer or whether RNAs can bind individual Hfq subunits and binding coincides with hexamerization of Hfq monomers as well as which other factors are involved in this process.

The presented manuscript builds on a previous study (Argaman et al., PNAS, 2012) from the authors, where the Altuvia lab reported that the RelA protein stimulates regulation by the small RNA RyhB by acting as a functional partner for the RNA chaperone Hfq. Their previous study suggested that RelA facilitates binding of Hfq to RyhB and that RelA enables low and ineffective concentrations of Hfq to bind RNA by promoting hexamerization of Hfq.

Here, using a set of in-vivo reporter gene assays of diverse RelA mutants as well as in vitro biochemical interaction studies among RelA, Hfq and selected RNAs, they now further study the underlying molecular basis of RelA mediated stimulation of Hfq function. Based on analysis of RelA mutant strains including truncations or point mutations, they demonstrate that the N-terminus of RelA is required for promoting RyhB-mediated target (*sodA/sdhC*) regulation and induces Hfq oligomerization and that the effect of RelA on RyhB-mediated post-transcriptional activity is independent of RelA mediated (p)ppGpp production. Furthermore, they demonstrate that Hfq co-immunoprecipitates with RelA in a RNA-dependent manner. Using a variety of in vitro gel-shifts and binding studies as well as structure probing experiments, they further demonstrate that RelA can directly bind to SD-like (GGAG) sequences in RyhB sRNA and *sodA* mRNA and that this binding of RelA to GGAG motifs in RNA is required to promote Hfq oligomerization and stabilizes RNA-Hfq complexes. Hereby, RelA binding stabilizes an initial complex of a Hfq monomer and RNA and facilitates binding of additional Hfq subunits. Finally, the authors demonstrate that binding to SD-like sequences in RNAs by RelA and its catalytic function on (p)ppGpp synthesis are mutually exclusive functions and they show that also the function of other RNAs with GGAG motifs (e.g. *OxyS*) is affected by RelA.

Overall, I think this study comprises very solid data that support the drawn conclusions and advances our knowledge on the molecular mechanisms of post-transcriptional regulation and assembly of RNP complexes in prokaryotes. I just have some comments for further clarifications.

Major comments:

-Several Figures miss statistical analyses and representation of significance of observed changes, e.g. calculation of p-values in Figures 1A, 6C-E, S1, S11. Also information about sample size and whether error bars indicate SD/SEM in the Figure legends is missing. These should be added.

-Figure 1A & S1: Have the authors checked (just as control) whether RyhB shows the same expression (e.g. on a Northern blot) in the different strains with the RelA mutant variants?

Minor comments:

-If I understand correctly, once RelA binds to an Hfq-RNA subunit and induces hexamerization, RelA is released from the complex. Thus, there is no stable association of RelA to the Hfq hexamer bound to RNA, right? Maybe try to clarify.

-The discussion appears very long to me. It could be streamlined by removing redundancies with the results section.

-Figure 6: The order of the panel labels E and D should be exchanged in the Figure (to fit the text on page 12-13). And please change the label of y-axis: NadE-lacZ -> nadE-lacZ

-Figure 7: I think the representation of SD-like sequences by a music note is not very intuitive. Maybe change to a more commonly used representation (e.g. box with label SD-like). Moreover, I would also find it helpful, if the sRNA and mRNA are labeled at least once in the Figure.

Reviewer #2 (Remarks to the Author):

Altuvia and coworkers follow-up on their previous paper (Argaman et al, 2012), showing that RelA, the primary ppGpp synthase in *E. coli*, was necessary for optimal Hfq-dependent regulation by the small RNA RyhB by increasing the levels of Hfq hexamers. In the current work, they look at this in more detail. They identify RelA binding motifs on the relevant RNAs, identify RelA mutants deficient in this binding but not in ppGpp synthesis, and demonstrate that the RNAs serve as a link for RelA and Hfq. Surprisingly, they find that single subunits of Hfq can bind RNA as well, contrary to previous findings, and that RelA helps the maturation of the Hfq hexamer from this initial step. The basic story is intriguing and will be of general interest. However, some of the data needs quantification and clarification, and because the *in vivo* work is done with multicopy RNAs and proteins, it is not always clear how this will operate under more physiological conditions.

1. Figure S1, 6: The separation of function in RelA is a critical part of this work. The Q264E allele is described as enabling *sdhC* regulation, although the results show significantly less regulation than WT. It seems important to also test this with *sodA* regulation, which has a significantly stronger regulation in the first place. This should be added to Figure 1.
2. Figure 1C: Please clarify what is being seen here. What are the high MW bands in lanes 2 and 3? Are the hexamers expected for Hfq not seen? Why are tetramers the diagnostic species here?
3. Figure S3: These plates could be better labeled. What is the third plate, some parts of which are not labeled on the side? With respect to separation of function, this is a very quantitative test for ppGpp, done with overproduced protein (correct?). Have any of the RelA alleles that block Hfq assembly function been introduced into the chromosomal copy of *relA* and tested for function?
4. Figure 2: How much is RelA overproduced in strains upon induction, for this co-immunoprecipitation? Is that necessary for this interaction? What is the fraction of Hfq that is interacting? Is the assumption that the assembled hexamer there would not be an interaction any more?
5. Figure 2B, Figure S4: Text suggests that Figure S4 uses truncated RNA residues, but figure legend does not clarify what these pieces are.
6. Can any quantitation be provided for efficiency of binding of RNA to RelA in Fig. 2C? The RelA-*sodA* is particularly low here. What happens with non-target RNAs?
7. P. 8, top lines: Clarify what the higher molecular weight complexes that are mentioned here? Is this a reference to the modest decrease in RyhB binding in lane 4 of Fig. 2B? Is it clear such complexes are the same as being incapable of RNA binding, as the text seems to say here?
8. While the authors identify GGAGA (in *sodA*) and GGAAGA (in RyhB) for RelA binding by the protection assays in Figure 3, and suggest that these sequences are SD-like, could it be that G-rich sequences are being bound? The changes made that abolish binding are rather drastic; were less

drastic changes tried? Are there any related sequences elsewhere in these RNAs that are not bound? Are these sequences in regions of the RNAs that are particularly accessible?

9. Figure 4: Quantitation of the bands or differences in bands (tetramer and dimer?) as function of RNA and RelA are particularly important here, since it is this data on which the basic model is to a large degree based. Reproducibility also needs to be documented. Why are higher MW oligomers seen without RNA (or with) in A and B, but much less prominent in C or D (without RNA, so should be the same). Given that RyhB seems to give better binding to RelA than *sodA*, why is the effect on Hfq multimerization less striking?

10. Figure 5 and model of binding: A number of issues are unclear for this data and its discussion in the text.

a. What is the evidence that RyhB binding to the Hfq monomer is using the same surfaces of Hfq as has been defined in the crystals or hexamer binding? For fig. 5B and D, distal site mutants that presumably should not bind *sodA* as well seem to bind it at the higher concentrations (assembled Hfq) (Fig. 5B, 5D, lane 9). In 5C, very little binding of RyhB is seen with D9A, but this proximal mutant is not really defective for sRNA binding in most tests.

b. Why were RelA concentrations so high in the experiments in 5B-5D? Where is RelA in these gels?

c. Why is Fig. S9 not discussed until significantly later in the paper (in the discussion)? It would have been useful to have it when Figure 5 is discussed. When it is discussed, it seems like the authors have ad hoc explanations for whatever does not fit – G29A binding of RyhB not stimulated by high RelA suggests competition, but as far as I can see, same experiment was not tried for K56A and D9A. How about for *sodA* with this distal site mutants at lower RelA?

11. Fig. S10: (missing S on line 16, p. 14): Why is this figure only presented in the discussion? It is data, presumably. This is another figure where either quantitation or further controls are needed to make the conclusions fully convincing. The authors highlight some multimeric bands as appearing in a pattern that fits their model, but exactly what these multimers are is not discussed, and the pattern is not as clear as suggested by the text. Panel A is the clearest. For panel B, the Hfq bands are decreased in lane 12. Why? Panel C, it is difficult to tell what is going on (why is lane 9 down, and what is happening in lane 12?). In Panel D, difference between distal and proximal RNAs in terms of Hfq bands is extremely modest, if there at all.

12. Fig. S11: Again, this should not be in the discussion.

a. Were these levels of RNA in the coIP compared at all to total RNA before IP? CoIP is with RelA? That is only said in the discussion, not in the figure legend.

b. I do not follow the results at all – *del hfq* has no effect on RyhB, McaS, or others? That seems extremely unlikely. Why isn't the usual decrease in levels of these RNAs seen?

c. Are the sRNAs expressed from plasmids or from the chromosome? MgrR seems to be abundant after IP, but doesn't care about the C289Y mutation. What is the interpretation of that?

Reviewer #3 (Remarks to the Author):

The manuscript by Basu and coworkers deals with a very interesting topic. How the ppGpp synthesizing protein RelA could stimulate Hfq activity in Gram negative bacteria. The paper is based on an earlier study by the co-workers (Argaman et al., 2012). Through a series of experiments, the authors suggest that RelA can stabilize the interaction between Hfq and GGAG (SD-sequence) carrying sRNAs, eventually allowing hexamerization of Hfq.

Although the topic is highly relevant, the paper is quite confusing to read, statistics are lacking and most importantly, the experiments do not fully support the conclusions. At its present form, I am unable to recommend publication.

Major comments:

1. For in vitro experiments, the concentration of RelA varies dramatically between the different experiments (ranging from 25 nM to 1000 nM). This makes me wondering of the biological relevance of the in vitro studies. What is the concentration of RelA in vivo? The authors refer to a paper (Pedersen and Kjeldgaard, 1977) where they estimate that there is around 100 RelA molecules/ cell.

What would such an amount reflect in RelA concentration? The difference in RelA concentration for the different experiments makes it hard to evaluate the potential impact of RelA for the process. This is especially troublesome since the concentration of RelA almost exclusively is larger than the concentration of Hfq, despite the latter being in much higher numbers intracellularly. Since the authors have RelA and Hfq antibodies as well as purified RelA and Hfq protein, it should be possible to calculate the intracellular concentrations of the different proteins at different conditions. Using fractionation methods, the authors should be able to determine the "free" versus "bound" RelA and Hfq concentrations. Using these numbers, the authors should repeat a couple of key in vitro experiments at relevant Hfq and RelA concentrations to validate their findings (Figure 1B, Figure 4, Figure 5 and Figure 6B).

2. The authors do not show any statistics, nor do they indicate how many times an experiment has been repeated. This is especially troublesome for some of the conclusions drawn (see below). The authors need to repeat (or indicate that they already done so) each experiment at least three times and provide relevant statistics.

3. Figure 1B and C. First, the authors are making conclusions from using two different RNAs (sodA and RyhB). Second, in Figure 1C, there is a larger migrating band in the -RelA and RelA C289Y lanes after 15 and 30 seconds that looks like Hfq multimerization. This is not discussed but could significantly change the conclusions drawn.

4. Figure 4. The conclusions drawn are far from supported. From the data presented it is very hard to draw any conclusions regarding multimerization of Hfq by sRNAs and/or RelA. The bands indicated by asterisks are very weak compared to controls. Why not using mutated sodA as in Figure 3 instead of sodA-ΔSD? Also, why using western blots to identify proteins in a cell-free experiments (same for Figure 1C)

5. Figure 5D, The interpretation of the data is very confusing. The authors are completely ignoring other stronger bands than the ones being highlighted by asterisks. What are these bands and why are they largely missing when HfqWT and HfqI30D are added simultaneously? Besides that, it is more relevant to compare lanes 14-17 with lane 7 since the concentration of Hfq is the same in these lanes.

6. Figure 6 A and B. It is very hard to make any conclusions from the gels presented if only done once. For instance, differentiating between a conversion of 71 to 46 % from 71 to 57 %, saying that the latter was "much less affected by the RNA" is a large over interpretation.

7. The Methods section is scarce on information. Two examples: No references for the antibodies used in this study are shown and no concentration of the sRNAs used in Figure 6B is given. Actually, it is very hard to find the concentration of RNA being used in any of the experiments presented.

8. The Figure legends are showing very little information of what is shown in the figure and how the experiments were conducted (e.g. concentration of RNA) but very much conclusions drawn by the authors.

9. In their previous paper (Argaman et al 2012), the authors could show that Hfq and RelA cosedimented with the ribosomes. Also, RyhB crosslinked to Hfq as well as ribosomal protein S1. This makes me wonder if the ribosome is important/required for the RelA activity on Hfq? Is there a competition for RelA between the ribosome and the sRNA?

Minor comments:

10. Figure 2C, the gel shifts look very strange. It seems that there is a 50% shift at 100 nM (as the authors indicate) but this shift is not further increasing. Rather, the amount of unbound RyhB remain the same even at very high concentrations of RelA. What is happening at RelA concentrations below 100 nM? Also, an upper band (possibly the edge of the well) show a decreasing amount of intensity, is there a lot of complex not entering the gel?

11. In Figure 5A, why is there a RyhB:Hfq band in the 0 minute fraction (lane 4) without any crosslinking?

12. Figure 6D and E, there is a mixup of panels.

January 7, 2021

Minju Ha, PhD
Associate Editor
Nature Communications
Re: Revised manuscript NCOMMS-20-32106A-Z

Dear Dr. Ha,

We wish to thank all the referees for their useful suggestions. To address the referee's comments, we carried out additional experiments. The main manuscript now includes new data. Below we present a point-by-point response to their comments.

Reviewer #1 (Remarks to the Author):

Major comments:

Several Figures miss statistical analyses and representation of significance of observed changes, e.g., calculation of p-values in Figures 1A, 6C-E, S1, S11. Also, information about sample size and whether error bars indicate SD/SEM in the Figure legends is missing. These should be added.

Reply: The revised manuscript now includes statistical analysis and representation of the significance of the observed changes.

Figure 1A & S1: Have the authors checked (just as control) whether RyhB shows the same expression (e.g. on a Northern blot) in the different strains with the RelA mutant variants?

Reply: qRT analysis of RyhB in wild type and RelA mutants shows that the levels of RyhB are approximately 2-fold higher in *relA* wild type than in *relA* mutants, indicating that RelA-sRNA binding allows moderate stabilization of the associated RNAs (presented in Supplementary Fig. 14c).

Minor comments:

If I understand correctly, once RelA binds to an Hfq-RNA subunit and induces hexamerization, RelA is released from the complex. Thus, there is no stable association of RelA to the Hfq hexamer bound to RNA, right? Maybe try to clarify.

Reply: Based on our data we propose that once RelA binds to Hfq-RNA complex and stabilizes it, to enable formation of an Hfq hexamer, RelA is released. The proposed mechanism assumes that RelA is effective in sub-stoichiometric amounts relative to Hfq, which in turn corresponds to the previously reported low intracellular concentration of RelA (Li et al., Cell 2014). This is now clarified in the discussion page 15.

The discussion appears very long to me. It could be streamlined by removing redundancies with the results section.

Reply: As per this comment we shortened the discussion by removing redundancies with the results section.

Figure 6: The order of the panel labels E and D should be exchanged in the Figure (to fit the text on page 12-13). And please change the label of y-axis: NadE-lacZ -> nadE-lacZ

Reply: Corrected

Figure 7: I think the representation of SD-like sequences by a music note is not very intuitive. Maybe change to a more commonly used representation (e.g. box with label SD-like). Moreover, I would also find it helpful, if the sRNA and mRNA are labeled at least once in the Figure.

Reply: Corrected as suggested

Reviewer #2 (Remarks to the Author):

Altuvia and coworkers follow-up on their previous paper (Argaman et al, 2012), showing that RelA, the primary ppGpp synthase in *E. coli*, was necessary for optimal Hfq-dependent regulation by the small RNA RyhB by increasing the levels of Hfq hexamers. In the current work, they look at this in more detail. They identify RelA binding motifs on the relevant RNAs, identify RelA mutants deficient in this binding but not in ppGpp synthesis, and demonstrate that the RNAs serve as a link for RelA and Hfq. Surprisingly, they find that single subunits of Hfq can bind RNA as well, contrary to previous findings, and that RelA helps the maturation of the Hfq hexamer from this initial step. The basic story is intriguing and will be of general interest. However, some of the data needs quantification and clarification, and because the *in vivo* work is done with multicopy RNAs and proteins, it is not always clear how this will operate under more physiological conditions.

1. Figure S1, 6: The separation of function in RelA is a critical part of this work. The Q264E allele is described as enabling *sdhC* regulation, although the results show significantly less regulation than WT. It seems important to also test this with *sodA* regulation, which has a significantly stronger regulation in the first place. This should be added to Figure 1.

Reply: As per this comment we tested the effect of Q264E mutant on *sodA* target. The results presented in Fig. 1a and in Supplementary Fig. 1 show that RelA:Q264E enables repression of *sodA-lacZ* and *sdhC-lacZ* by RyhB, further confirming that (p)ppGpp production was not associated with RelA regulation of basepairing RNAs. We edited the text accordingly (page 6).

2. Figure 1C: Please clarify what is being seen here. What are the high MW bands in lanes 2 and 3? Are the hexamers expected for Hfq not seen?

Reply: We repeated the experiment presented in Fig. 1c adding a sample of 70S ribosomes prepared from wild type cells for which we found that in contrast to purified Hfq protein that exhibits high molecular weight oligomers, 70S preparation presents Hfq lower oligomeric bands including dimers, tetramers and hexamers. Similar to the previous experiment this new gel exhibits an increase in the levels of Hfq dimers, tetramers and hexamers, when incubated with RelA. The oligomerization pattern of RelA:C298Y mutant is similar to the pattern detected in the absence of RelA.

Why are tetramers the diagnostic species here?

Reply: Our data show that RelA binds and stabilizes a binary complex of RNA bound to Hfq monomer enabling the attachment of additional subunits to form hexamers. To follow the Hfq assembly process we observe the levels of lower oligomeric bands including dimers, tetramers as well as hexamers.

3. Figure S3: These plates could be better labeled. What is the third plate, some parts of which are not labeled on the side? With respect to separation of function, this is a very quantitative test for ppGpp, done with overproduced protein (correct?).

Reply: To distinguish between the RelA synthetase function from its RNA binding function we constructed three RelA mutants at position Q264 based on RelA_{Seq} of *Streptococcus dysgalactiae*. Changing glutamate (Q) to glutamic acid (E) rendered RelA inactive as synthetase, whereas changing it to alanine (A) or to asparagine (N) had no effect on the activity of RelA as a synthetase. These mutants are now described in Supplementary Fig. 3 (see also the legend). The plating assays were carried out with low copy plasmids (P15A) encoding the various *relA* alleles.

Have any of the RelA alleles that block Hfq assembly function been introduced into the chromosomal copy of *relA* and tested for function?

Reply: *relA*:C289Y allele was introduced into the chromosome. This chromosomally encoded mutant allele was examined in vivo for its ability to produce ppGpp (Fig. 6a) as well as to bind RNA in Co-IP using α RelA antibody. (Fig. 2a; Supplementary Fig.16).

4. Figure 2: How much is RelA overproduced in strains upon induction, for this co-immunoprecipitation? Is that necessary for this interaction?

Reply: Co-IP experiments were carried out with chromosomally encoded *relA* wild type and *relA*:C289Y alleles. The chromosomal levels of RelA suffice for RelA-RNA Hfq interaction.

What is the fraction of Hfq that is interacting? Is the assumption that the assembled hexamer there would not be an interaction any more?

Reply: We assume that once the hexamer is formed it will no longer interact with RelA.

To estimate the fraction of Hfq that can interact with RelA in cells lysates, we examined samples collected before and after Co-IP. Purified Hfq was used as control. Band intensities of Hfq monomers, dimers and dodecamers were measured using ImageLab software and the number of subunits in each form was calculated as compared to the purified Hfq. We estimate that about 50% (± 5) of Hfq can interact with RelA in cell lysates (Supplementary Fig. 16a). Given that the number of molecules of RelA and Hfq per cell before Co-IP were estimated to be 4,000 and 15,360, respectively, the ratio of the Hfq fraction (7680 molecules) interacting with RelA (4,000 molecules) is around two, suggesting that one molecule of RelA interacts with two molecules (monomers) of Hfq (Supplementary Fig. 16b). Supplementary Fig. 16 is discussed in the discussion section (pages 15 and 16).

5. Figure 2B, Figure S4: Text suggests that Figure S4 uses truncated RNA residues, but figure legend does not clarify what these pieces are.

Reply: In these experiments labelled RyhB and *sodA* RNAs were incubated with RelA followed by UV crosslinking. Supplementary Fig. 4 presents SDS-PAGE analysis of RelA-RNA binding

products after treatment with RNase A to remove unprotected RNA residues. The estimated MW of the trimmed binding products is ~ 85 kDa. Fig. 2b present the same binding products prior to exposure to RNase A. The estimated MW of these products is ~ 110 kDa.

This is now explained in the legend to Supplementary Fig. 4, and in the text (page 7)

6. Can any quantitation be provided for efficiency of binding of RNA to RelA in Fig. 2C? The RelA-*sodA* is particularly low here.

Reply: We quantitated the efficiency of binding of RNA to RelA. The revised manuscript includes mean and SD based on two sets of experiments. We estimate that approximately 30 to 40 % of RyhB is bound by RelA. The affinity of RelA for *sodA* is much weaker. We corrected the text accordingly.

What happens with non-target RNAs?

Reply: Just for the referees we provide a figure showing that RelA does not bind *nadE*, a non-target RNA.

7. P. 8, top lines: Clarify what the higher molecular weight complexes that are mentioned here? Is this a reference to the modest decrease in RyhB binding in lane 4 of Fig. 2B? Is it clear such complexes are the same as being incapable of RNA binding, as the text seems to say here?

Reply: We thank the referee for his comment, indeed, we failed to explain this observation. We suspect the decrease is due to the formation of high molecular weight RelA complexes incapable of RNA binding. These complexes are likely formed at high concentrations of RelA due to UV-mediated increased covalent cross linking of aromatic residues (Francesco Itri et al. Cell Mol Life Sci. 2016). We corrected the text and added this reference (page 8).

8. While the authors identify GGAGA (in *sodA*) and GGAAGA (in RyhB) for RelA binding by the protection assays in Figure 3, and suggest that these sequences are SD-like, could it be that G-rich sequences are being bound? The changes made that abolish binding are rather drastic; were less drastic changes tried? Are there any related sequences elsewhere in these RNAs that are not bound? Are these sequences in regions of the RNAs that are particularly accessible?

Reply: Our foot-printing experiments showed that RelA protected the sequence GGAGA in both *sodA* and RyhB. RyhB also consists of a variation of this sequence (GGAAGA) but RelA did not protect this site, suggesting that RelA recognizes more than just G rich sequences. GGAGA site of RyhB seems to be accessible.

9. Figure 4: Quantitation of the bands or differences in bands (tetramer and dimer?) as function of RNA and RelA are particularly important here, since it is this data on which the basic model is to a large degree based. Reproducibility also needs to be documented.

Reply: In the revised manuscript, Fig. 4 includes the MW marker that was originally in the gel alongside the other samples. By comparing this pattern with the Hfq oligomeric pattern observed in the 70S sample (new Fig. 1c) we re-denoted the oligomeric forms. Quantitation of the bands is provided in Supplementary Fig. 6. For reproducibility, a second set of this experiment is shown in Supplementary Fig. 7

Why are higher MW oligomers seen without RNA (or with) in A and B, but much less prominent in C or D (without RNA, so should be the same). Given that RyhB seems to give better binding to RelA than *sodA*, why is the effect on Hfq multimerization less striking?

Reply: The differences pointed out by the reviewer, (a,b vs. c,d) are intrinsic to assays; as can be seen not only the higher MW oligomers are underscored in a and b but the monomers are intensified too. To be able to compare between with and without RNA each experiment was analyzed in separate gels. ECL is a very sensitive assay and although showing similar changes in the oligomeric patterns, the intensities vary. Thus, a second set of Fig. 4c and d with its own quantitation is provided in Supplementary Fig. 7.

10. Figure 5 and model of binding: A number of issues are unclear for this data and its discussion in the text.

a. What is the evidence that RyhB binding to the Hfq monomer is using the same surfaces of Hfq as has been defined in the crystals or hexamer binding?

Reply: In Fig. 5 we show that Hfq:D9A, a proximal face mutant is unable to bind RyhB, a proximal face RNA. By providing a minimal amount of wild type Hfq to initiate RyhB binding to Hfq monomer we lay the base foundation for formation of mixed oligomeric complexes.

For fig. 5B and D, distal site mutants that presumably should not bind *sodA* as well seem to bind it at the higher concentrations (assembled Hfq) (Fig. 5B, 5D, lane 9). In 5C, very little binding of RyhB is seen with D9A, but this proximal mutant is not really defective for sRNA binding in most tests.

Reply: The Hfq mutants presented in this study are not totally null, showing some activity of RNA binding when at high concentrations. However as oppose to wild type, low concentrations of these mutants (5nM) do not undergo multimerization in the presence of RelA, unless presented by wild type Hfq monomer bound to RNA. As for D9A; The Hfq D9A allele was first isolated for its failure to support negative regulation of *sdhC* by RyhB sRNA (Panja et al., JMB 2015).

b. Why were RelA concentrations so high in the experiments in 5B-5D? Where is RelA in these gels?

Reply: In the first Gel mobility shift experiment (Fig.1b) we used three different concentrations of RelA (50, 250 and 500 nM). Based on that data we decided to use 200 nM of RelA in EMSA assays.

c. Why is Fig. S9 not discussed until significantly later in the paper (in the discussion)? It would have been useful to have it when Figure 5 is discussed. When it is discussed, it seems like the authors have ad hoc explanations for whatever does not fit – G29A binding of RyhB not stimulated by high RelA suggests competition, but as far as I can see, same experiment was not tried for K56A and D9A. How about for *sodA* with this distal site mutants at lower RelA?

Reply: As per this referee suggestion we now present Supplementary Fig. 9 in the results (page 11-12). The revised manuscript also includes mobility shift data of K56A and D9A Hfq mutants incubated with *sodA* RNA in the presence of low and high concentrations of RelA. The results show that only high concentrations of RelA can facilitate RNA binding to Hfq:K56A and Hfq:D9A Supplementary Fig. 9. The explanation for this phenomenon is provided in the text (page 11-12)

11. Fig. S10: (missing S on line 16, p. 14): Why is this figure only presented in the discussion? It is data, presumably. This is another figure where either quantitation or further controls are needed to make the conclusions fully convincing. The authors highlight some multimeric bands as appearing in a pattern that fits their model, but exactly what these multimers are is not discussed, and the pattern is not as clear as suggested by the text. Panel A is the clearest. For panel B, the Hfq bands are decreased in lane 12. Why? Panel C, it is difficult to tell what is going on (why is lane 9 down,

and what is happening in lane 12?). In Panel D, difference between distal and proximal RNAs in terms of Hfq bands is extremely modest, if there at all.

Reply: In the revised manuscript we included the molecular weight markers that were originally in the gels. In addition, we repeated the experiment presented in Fig. 1c adding a sample of 70S ribosomes prepared from wild type cells for which we found that in contrast to purified Hfq protein that exhibits high molecular weight oligomers, 70S preparation presents Hfq lower oligomeric bands including dimers, tetramers and hexamers. Accordingly, Hfq multimers formed were denoted and quantitated. For reproducibility, we decided to provide a second set of the experiments presented in this Figure. This set shows similar changes in the oligomeric patterns, however, given that ECL is a very sensitive assay depending on the exposure duration, the intensities of every set is provided with its own quantitation (Supplementary Fig. 10 11 and 12)

Indeed, lane 12 in panel B was decreased due to some technical issues. Thus, we repeated the experiment. The revised manuscript includes the new experiment. As for the results in panels c and d, as observed in these panels, in contrast to wild type (see Fig. 4) and the proximal face mutants (in this figure), the distal face mutants I30D and G29A tend to accumulate high multimers even in the absence of any RNA. This phenomenon is enhanced with protein cross linking; lane 9 in panel c for example exhibits a very high molecular weight complex that got stuck on top. However, because RelA helps with the multimerization process, we set to observe the steps of multimerization by following the formation of low molecular weight multimers *i.e.*, dimers, tetramers and pentamers. The quantitation shows that in the presence of RelA, the combinations K56A/*sodA*, D9A/*sodA*, G29A/RyhB, and I30D/RyhB lead to an increase in the levels of those multimers.

12. Fig. S11: Again, this should not be in the discussion.

Reply: As per this referee comment Supplementary Fig. 9 and 10 were included in the results. We thought that the data presented in former Fig.11, now Supplementary Fig. 17 is a better fit in the discussion.

a. Were these levels of RNA in the coIP compared at all to total RNA before IP? CoIP is with RelA? That is only said in the discussion, not in the figure legend.

Reply: In the revised manuscript we provide RT-PCR analyses of the different RNAs before and after Co-IP (new Supplementary Fig. 17). As a rule, both GGAG sRNAs and those lacking the site exhibit similar RNA levels before Co-IP. After Co-IP, GGAG sRNA levels are significantly higher in wild type RelA cells compared to the levels detected in RelA:C289Y mutant. The levels of sRNAs that lack GGAG are similar in both wild type and RelA:C289Y mutant, while GGAG mRNA targets exhibit a somewhat lower level in RelA mutant. Interestingly, before Co-IP, the levels of both *sodA* and *sdhC* RyhB targets are lower in wild type RelA cells compared to RelA:C289Y, probably due to RelA mediated repression by RyhB.

Co-immunoprecipitation was carried out with α RelA antibody; this information was mentioned in the legend to Fig. 2, in Methods and in the Discussion as said. In the revised manuscript we make sure that this critical information is indeed clear.

b. I do not follow the results at all – del hfq has no effect on RyhB, McaS, or others? That seems extremely unlikely. Why isn't the usual decrease in levels of these RNAs seen?

Reply: The assays of RT-PCR carried out on RNAs before and after Co-IP demonstrate that the sRNA levels of RyhB, McaS, MgrR and MicC decrease in Δhfq mutant. In contrast, the stability of SraC sRNA and *sodA* and *sdhC* mRNAs are not affected by Hfq.

c. Are the sRNAs expressed from plasmids or from the chromosome? MgrR seems to be abundant after IP, but doesn't care about the C289Y mutation. What is the interpretation of that?

Reply: The genes expressing the sRNAs and mRNAs investigated in Co-IP are chromosomally encoded. MgrR doesn't care about RelA:C289Y mutant because unlike RyhB, MgrR does not carry the RelA binding site GGAG.

Reviewer #3 (Remarks to the Author):

Major comments:

1. For in vitro experiments, the concentration of RelA varies dramatically between the different experiments (ranging from 25 nM to 1000 nM). This makes me wondering of the biological relevance of the in vitro studies. What is the concentration of RelA in vivo? The authors refer to a paper (Pedersen and Kjeldgaard, 1977) where they estimate that there is around 100 RelA molecules/ cell. What would such an amount reflect in RelA concentration? The difference in RelA concentration for the different experiments makes it hard to evaluate the potential impact of RelA for the process. This is especially troublesome since the concentration of RelA almost exclusively is larger than the concentration of Hfq, despite the latter being in much higher numbers intracellularly. Since the authors have RelA and Hfq antibodies as well as purified RelA and Hfq protein, it should be possible to calculate the intracellular concentrations of the different proteins at different conditions. Using fractionation methods, the authors should be able to determine the "free" versus "bound" RelA and Hfq concentrations. Using these numbers, the authors should repeat a couple of key in vitro experiments at relevant Hfq and RelA concentrations to validate their findings (Figure 1B, Figure 4, Figure 5 and Figure 6B).

Reply: Per your comment we revisited RelA concentrations. In a latter study, by quantifying absolute protein synthesis rates, Li et al (Cell 2014) identified 213, 356 and 358 RelA molecules per generation for three growth conditions in minimal medium at exponential phase, corresponding to RelA concentrations of 200 nM - 330 nM. Accordingly, the concentrations we used to investigate Hfq assembly or RelA stabilization of RNA bound to one Hfq monomer (25 nM of RelA) were within the physiological concentration range. Gel mobility shift experiments in which we investigated Hfq-RNA and RelA-RNA binding in native gels or in SDS-PAGE after cross linking were carried out using a range of RelA concentrations (100 to 500 nM), however binding was detected using < 250 nM of RelA.

To assess the concentrations of RelA and Hfq under the growth conditions used in this study we determined the intracellular levels of RelA and Hfq. Our assessment of the concentrations of chromosomally and plasmid encoded RelA in exponential phase is 518 ± 150 nM and 2 ± 1 μ M, respectively (Supplementary Fig. 15). Our estimation of Hfq monomer concentration carried out under the same conditions show that *relA+* and *relA-* strains harbor approximately 4.5 μ M and 2.5 μ M, respectively (Supplementary Fig. 15). We also measured the stationary phase intracellular concentrations of RelA (2.2 ± 0.15 μ M) and Hfq (8.5 ± 3 μ M). The increase in RelA expression in stationary phase is due to the stationary phase dependent P2 promoter of RelA (Nakagawa et al., 2006; Genes Genet Syst; Brown et al., 2014; Nature Communications). These measurements indicate that the levels of Hfq are 9 and 4-fold higher than the levels of RelA in exponential and stationary phase respectively. Yet, as discussed in the text (page 15) the absolute concentration of Hfq is not indicative of Hfq availability. Co-IP studies have revealed thousands of Hfq-bound RNAs

and overexpression of Hfq-dependent sRNAs resulted in the sequestration of Hfq and thus in Hfq depletion. Here, we show that RelA enables binding of RNAs by otherwise ineffective amounts of Hfq *in vitro*, and facilitates Hfq mediated basepairing regulation of specific sRNA/mRNA pairs *in vivo*, indicating that under specific conditions and/or environments, Hfq availability is inadequate.

In addition, based on our Co-IP studies we estimate that about 50 % (± 5) of Hfq can interact with RelA in cell lysates (Supplementary Fig. 16a). Given that the number of molecules of RelA and Hfq per cell before Co-IP were estimated to be 4,000 and 15,360, respectively, the ratio of the Hfq fraction (7680 molecules) interacting with RelA (4,000 molecules) is around two, suggesting that one molecule of RelA interacts with two molecules (monomers) of Hfq (Supplementary Fig.16b). Supplementary Fig. 16 is discussed in the discussion section (page 15 and 16).

2. The authors do not show any statistics, nor do they indicate how many times an experiment has been repeated. This is especially troublesome for some of the conclusions drawn (see below). The authors need to repeat (or indicate that they already done so) each experiment at least three times and provide relevant statistics.

Reply: The revised manuscript includes statistics and details of the number of biological samples used in each experiment wherever needed. The legends were edited accordingly.

3. Figure 1B and C. First, the authors are making conclusions from using two different RNAs (sodA and RyhB).

Reply: We use and characterize the binding of both RyhB and SodA throughout the study. As can be seen later in the study we based the conclusions by characterizing both RNAs.

Second, in Figure 1C, there is a larger migrating band in the -RelA and RelA:C298Y lanes after 15 and 30 seconds that looks like Hfq multimerization. This is not discussed but could significantly change the conclusions drawn.

Reply: We repeated the experiment presented in Fig. 1c adding a sample of 70S ribosomes prepared from wild type cells for which we found that in contrast to purified Hfq protein that exhibits high molecular weight oligomers, 70S preparation presents Hfq lower oligomeric bands including dimers, tetramers and hexamers. Similar to the previous experiment this new gel exhibits an increase in the levels of Hfq dimers, tetramers and hexamers, when incubated with RelA. The oligomerization pattern of RelA:C298Y mutant is similar the pattern detected in the absence of RelA. The large migrating bands of Hfq in the absence of RelA are in the range of > 180 kDa likely representing inactive Hfq aggregates due to cross linking.

Our data show that RelA binds and stabilizes a binary complex of RNA bound to Hfq monomer enabling the attachment of additional subunits to form hexamers. To follow the Hfq assembly process we observe the levels of lower oligomeric bands including dimers tetramers as well as hexamers.

4. Figure 4. The conclusions drawn are far from supported. From the data presented it is very hard to draw any conclusions regarding multimerization of Hfq by sRNAs and/or RelA. The bands indicated by asterisks are very weak compared to controls. Why not using mutated sodA as in Figure 3 instead of sodA- Δ SD?

Reply: In the revised manuscript, Fig. 4 includes the MW marker that was originally in the gel alongside the other samples. By comparing this pattern with the Hfq oligomeric pattern observed in the 70S sample (new Fig.1c) we re-denoted the oligomeric forms. Quantitation of the bands is

provided in Supplementary Fig. 6. The quantitation presented in this Figure shows that the levels of higher molecular weight forms including tetramers and pentamers increased when Hfq was incubated with wild type RyhB and *sodA*, in the presence of RelA. For reproducibility, we provide a second set of panel b and d with its own quantitation (Supplementary Fig.7). In this set we used as suggested *sodA* mutant in which the GGAGA sequence was replaced by ACUCU. This set shows similar changes in the oligomeric patterns.

Also, why using western blots to identify proteins in a cell-free experiments (same for Figure 1C)

Hfq tends to form high molecular weight complexes when present at high concentrations. However, to detect the effect of RelA on Hfq assembly we had to use very low concentrations of Hfq. Thus, we chose to use antibodies as opposed to direct staining.

5. Figure 5D, The interpretation of the data is very confusing. The authors are completely ignoring other stronger bands than the ones being highlighted by asterisks. What are these bands and why are they largely missing when HfqWT and HfqI30D are added simultaneously? Besides that, it is more relevant to compare lanes 14-17 with lane 7 since the concentration of Hfq is the same in these lanes.

Reply: Lanes 14-17 comprise of mixed oligomeric complexes which are a combination of Hfq wild type and Hfq:I30D. Since Hfq:I30D forms a characteristic band pattern with the RNA we cannot compare it only to lane 7 that carries only wild type Hfq but with both lanes 7 and 9. Concomitantly with an increase in the concentration of the wild type from lanes 14 to 17 we observe a shift in the band pattern of this complex that resembles the wild type pattern. As for the higher bands, when at high concentrations both Hfq wild type and I30D form additional secondary complexes. These bands are mostly absent in lanes 14-17 except for the secondary complex detected in lane 3 and 7 of wild type Hfq. We marked the additional bands in lanes 3 7 and 17 as explained here.

6. Figure 6 A and B. It is very hard to make any conclusions from the gels presented if only done once. For instance, differentiating between a conversion of 71 to 46 % from 71 to 57 %, saying that the latter was “much less affected by the RNA” is a large over interpretation.

Reply: We repeated the quantitation of the amounts of ppGpp produced using the ImageLab software. We now present mean and standard deviation of 2 biological samples estimating ppGpp production in vivo and in vitro (Fig. 6a and b and Supplementary Fig. 13).

7. The Methods section is scarce on information. Two examples: No references for the antibodies used in this study are shown and no concentration of the sRNAs used in Figure 6B is given. Actually, it is very hard to find the concentration of RNA being used in any of the experiments presented.

Reply: We inspected our methods section and legends and filled in the missing information. The source of the antibodies is now described in section of western blotting.

8. The Figure legends are showing very little information of what is shown in the figure and how the experiments were conducted (e.g. concentration of RNA) but very much conclusions drawn by the authors.

Reply: We tried to fill in the missing information including RNA and protein concentrations

9. In their previous paper (Argaman et al 2012), the authors could show that Hfq and RelA cosedimented with the ribosomes. Also, RyhB crosslinked to Hfq as well as ribosomal protein S1. This makes me wonder if the ribosome is important/required for the RelA activity on Hfq? Is there a competition for RelA between the ribosome and the sRNA?

Reply: We suspect that the ribosome is an important environment that allows the implementation of RelA-RNA-Hfq regulation. The synthetase activity of RelA is induced by sensing uncharged tRNA upon amino acid starvation. This activity is reduced when GGAG sRNAs are present, indicating that these two functions are mutually exclusive. Having said that, our results showing that RelA of 70S extracts enables binding of labeled RNA by Hfq monomer indicate that the ribosomes provide a supporting environment for RelA-RNA-Hfq monomer binding, possibly by creating a micro-environment that holds the components in the correct position and structure. This is now discussed in the discussion section page 14.

Minor comments:

10. Figure 2C, the gel shifts look very strange. It seems that there is a 50% shift at 100 nM (as the authors indicate) but this shift is not further increasing. Rather, the amount of unbound RyhB remain the same even at very high concentrations of RelA. What is happening at RelA concentrations below 100 nM? Also, an upper band (possibly the edge of the well) show a decreasing amount of intensity, is there a lot of complex not entering the gel?

Reply: To clarify this issue we quantitated the efficiency of binding of RNA to RelA. The revised manuscript includes mean and SD based on two sets of experiments. We estimate that under these conditions RelA is capable of binding up to 30 - 40 % whereas the affinity of RelA for *sodA* is much weaker. We corrected the text accordingly.

Clearly, unlike Hfq which when at high concentrations binds most of the RNA, *in vitro* RNA binding by RelA is limited reaching a plateau of ~ 40%. As explained in the manuscript, since the *in vivo* complex of RNA bound by Hfq and RelA is sufficiently stable to be precipitated by RelA antibody, we suspect that RelA binding of RNA that is structurally modified by Hfq is more efficient. In the absence of the Hfq, the interaction of RelA with unaltered RNA is more elusive.

11. In Figure 5A, why is there a RyhB:Hfq band in the 0 minute fraction (lane 4) without any crosslinking?

Reply: The samples were exposed to UV for RNA-protein cross linking and only then to protein-protein crosslinking with 0.2% of glutaraldehyde. Time (min) indicates duration of protein cross-linking. We edited the legend text to clarify this point.

12. Figure 6D and E, there is a mixup of panels.

Corrected

We thank the referees for their comments and hope that the revised manuscript is now acceptable for publication.

Sincerely,

Shoshy Altuvia, Prof.

REVIEWER COMMENTS

Reviewer #1 (Remarks to the Author):

The authors have addressed all my previous comments in their revised manuscript.

Here are just two minor comments:

-p. 34, legend Fig. 1c: I think from the label in the gel/the figure legend it is not really clear what is „Hfq*“. I assume the 70S fraction. Please use consistent labels in Figure and legend.

-p. 35, legend Fig. 2c: ...experiments two for... -> remove „two“

Reviewer #2 (Remarks to the Author):

This revised version of the study of the intersection of RelA and Hfq in assembly of Hfq multimers, and their effects on sRNA-based regulation, is significantly clearer than previously. Quantitation of effects, added here, also adds to the story.

While the extent to which this RelA dependence for some regulation operates in vivo under physiological conditions and how it affects the hierarchy of sRNA regulation is not fully explored, the authors make a convincing case for a role of RelA, separate from its role in ppGpp synthesis, in promoting Hfq function via changes in multimerization. The results should be of significant interest in dissecting what Hfq does and how it does it.

Reviewer #3 (Remarks to the Author):

The manuscript by Basu and coworkers has been improved after the first submission and their results are interesting. However, I still have some major issues with experiments and conclusions made and cannot support publication in Nature Communications.

Major comments:

1. I much appreciate the effort by the authors to calculate the Hfq and RelA concentrations in the bacteria. With the new data, it appears that the concentrations of RelA used in most experiments are well within the intracellular range. However, the concentration of Hfq seems to be much higher intracellularly compared to the concentrations used in the experiments. In many of the experiments, the amount used is 5 nM which lies between 500-1000 times lower than the intracellular concentration. This is troublesome since higher concentrations of Hfq (100 nM and 250 nM) still can shift *sodA* and *RyhB* also in absence of RelA (Figure 1B and 5C). Thus, although many of the Hfq-proteins might still be occupied by binding other RNAs in the bacteria, I am still worried that the effects observed in the in vitro experiments does not mirror the in vivo situation.

Also, the concentration measurement of RelA and Hfq between Figure S15 and S16B does not match. In Figure S15, very much fewer bacteria were used to determine the concentrations of RelA and Hfq compared to Figure S16. For instance in Figure S15, approximately 1×10^5 RelA+ bacteria gave a RelA concentration of 1 nM whereas in Figure S16B, approximately 1×10^7 RelA+ bacteria were needed to get a similar concentration. What does this mean?

In Figure S15 and S16, the Western blot for RelA is very narrow. To fully evaluate the Co-IP studies done, it is required that the entire western blot is shown to make sure no other unrelated proteins also can be detected, and hence Co-IP'ed.

2. The results of adding purified ribosomes to the reactions is intriguing but unfortunately raises

several new questions. What I can see, neither the Methods section, the Results section nor the Figure Legends indicate anything of addition of ribosomes to the reactions. Neither can I identify any explanation of what source or amount of "purified 70S ribosomes" that were added. Since they were from WT cells, I presume they might carry RelA and Hfq as well? No section explains how they were purified. Also and maybe more importantly, did the authors add purified ribosomes to any of the other experiments? This is very important for the conclusions drawn in the paper. Nevertheless, addition of ribosomes to the reactions did not entirely remove the presence of multiprotein Hfq complexes (dodecamers?) in the -RelA and RelAC289Y reactions. A strange thing with the new Figure 1C is the overall low levels of Hfq in the -RelA and RelAC289Y lanes (lanes 1,2,3, 7 and 8). Can RelA protect Hfq from proteolysis?

3. In Figure S3, it is clear that a strain carrying RelAQ264E cannot grow in M9 minimal medium with 0.04% glucose, 0.4% glycerol and 0.1% arabinose. However, In Figure 1A and Figure S1, the same strain is able to grow in the same media (with 0.2% arabinose) and generate B-galactosidase expression. How is the strain able to grow?

4. In Figure 4 and S7, there is a poor reproducibility of the gels. In Figure 4D, very few Hfq multimers appear whereas most Hfq in Figure S7 is in a multimeric form (both in absence and presence of RyhB).

5. Although the authors provide some more statistics, it is far from sufficient. No detailed explanation of what type and settings of the the t-test they are using (only "unpaired"). Also, the use of asterisks are very confusing, sometimes a "*" indicates a P-value < 0.5 (Figure 6) whereas it also can mean > 0.1 (Figure S1). The same goes for the other asterisks. Generally, asterisks should only be shown for results showing a statistical significance (P-value < 0.05). Otherwise, the results should be named ns (non significant).

6. Also, most of the experiments have only been performed once which is not sufficient since reproducibility is an issue (e.g. point 4 above for Figure 4D and Figure S7). This is especially important for "key" figures like Figure 5A where the authors claim that RelA facilitates time-wise progression of a RyhB:Hfq interaction.

7. According to data in Figure S17, there is a very minute (close to none) effect of a RelAC289Y allele compared to RelAWT when determining the levels of sodA after Co-IP using RelA. However, as seen in Figure 2B and 3 the RelAC289Y mutant protein still binds much weaker to sodA compared to the WT protein. How can this be explained?

Minor comments:

8. Figure S3, why showing G251A, G251E and H354Y if not part of story?

9. Why is the RelA:SodA complex much larger than RelA:RyhB complex in Figure 2C despite these complexes having a similar size in Figure 2B?

10. Figure 5D, lanes 14-17 are still quite confusing, why is the shift lost in lane 16? Also, the brown asterisks sometimes show one complex or two complexes. In lane 7 for instance, a low Hfq concentration gives a large complex whereas such complex cannot be observed in lane 17. What does these results mean?

11. Figure 6B, despite claiming that in their answer to referees letter, the authors dis not repeat the experiment, or at least it is not written in the text.

February 11, 2021

Minju Ha, PhD
Associate Editor
Nature Communications
Re: Revised manuscript NCOMMS-20-32106B

Dear Dr. Ha,

We wish to thank the referees for their suggestions. We addressed the referee's comments point-by-point as described below.

Reviewer #1

The authors have addressed all my previous comments in their revised manuscript. Here are just two minor comments:

-p. 34, legend Fig. 1c: I think from the label in the gel/the figure legend it is not really clear what is „Hfq“. I assume the 70S fraction. Please use consistent labels in Figure and legend.
Reply: Fig.1C. Corrected as suggested

-p. 35, legend Fig. 2c: ...experiments two for... -> remove „two“
Reply: Legend Fig.2C. Corrected as suggested.

Reviewer #3

The manuscript by Basu and coworkers has been improved after the first submission and their results are interesting. However, I still have some major issues with experiments and conclusions made and cannot support publication in Nature Communications.

Major comments:

1. I much appreciate the effort by the authors to calculate the Hfq and RelA concentrations in the bacteria. With the new data, it appears that the concentrations of RelA used in most experiments are well within the intracellular range. However, the concentration of Hfq seems to be much higher intracellularly compared to the concentrations used in the experiments. In many of the experiments, the amount used is 5 nM which lies between 500-1000 times lower than the intracellular concentration. This is troublesome since higher concentrations of Hfq (100 nM and 250 nM) still can shift *sodA* and *RyhB* also in absence of RelA (Figure 1B and 5C). Thus, although many of the Hfq-proteins might still be occupied by binding other RNAs in the bacteria, I am still worried that the effects observed in the *in vitro* experiments does not mirror the *in vivo* situation.

Reply: It is common knowledge that the absolute concentration of Hfq is not indicative of Hfq availability. Hfq is engaged by thousands of RNAs *in vivo*. As mentioned in the discussion, Co-IP and RIL-Seq studies have shown that Hfq is bound by thousands of all sorts of RNAs. Moreover, overexpression of Hfq-dependent sRNAs resulted in the sequestration of Hfq and thus in Hfq depletion^{13,41-43}. Given that the majority of Hfq is unavailable for RNA binding *in vivo* and that the affinity of Hfq varies from RNA to RNA, we set to investigate whether RelA enables binding of RNAs by otherwise ineffective amounts of Hfq. As seen in Fig.1a and Supplementary Fig.1 our *in vivo* results demonstrating that RelA is essential for Hfq mediated basepairing regulation of specific

sRNA/mRNA pairs strongly confirm that under specific conditions and/or environments, RelA is mandatory to facilitate binding of RNA to the in vivo low levels of Hfq.

Also, the concentration measurement of RelA and Hfq between Figure S15 and S16B does not match. In Figure S15, very much fewer bacteria were used to determine the concentrations of RelA and Hfq compared to Figure S16. For instance in Figure S15, approximately 1×10^5 RelA+ bacteria gave a RelA concentration of 1 nM whereas in Figure S16B, approximately 1×10^7 RelA+ bacteria were needed to get a similar concentration. What does this mean?

In Figure S15 and S16, the Western blot for RelA is very narrow. To fully evaluate the Co-IP studies done, it is required that the entire western blot is shown to make sure no other unrelated proteins also can be detected, and hence Co-IP'ed.

Reply: In Supplementary Fig. 15, we present the intracellular concentrations of the proteins in **exponential phase**, whereas in Supplementary Fig. 16b we present the intracellular concentrations of the proteins in **stationary phase**. The differences in the protein concentrations vis-à-vis growth phases were elaborately discussed in the discussion section (page 15).

Also, the entire western blots are presented in the source data files along with the manuscript.

2. The results of adding purified ribosomes to the reactions is intriguing but unfortunately raises several new questions. What I can see, neither the Methods section, the Results section nor the Figure Legends indicate anything of addition of ribosomes to the reactions. Neither can I identify any explanation of what source or amount of “purified 70S ribosomes” that were added. Since they were from WT cells, I presume they might carry RelA and Hfq as well? No section explains how they were purified. Also and maybe more importantly, did the authors add purified ribosomes to any of the other experiments? This is very important for the conclusions drawn in the paper. Nevertheless, addition of ribosomes to the reactions did not entirely remove the presence of multiprotein Hfq complexes (dodecamers?) in the –RelA and RelAC289Y reactions. A strange thing with the new Figure 1C is the overall low levels of Hfq in the –RelA and RelAC289Y lanes (lanes 1,2,3, 7 and 8). Can RelA protect Hfq from proteolysis?

Reply: We would like to emphasize that **we never added 70S ribosomes to any of the reactions**. We used this sample as a molecular weight marker of the different Hfq forms. The use of 70S as a marker including the amount (4 μ g) were clearly explained in the figure legend of Fig 1c. The protocol for 70S preparation appears in Argaman et al., 2012 and in the legend to Fig. 1c. Also, Fig 1c is an in vitro experiment thus no proteolysis.

3. In Figure S3, it is clear that a strain carrying RelAQ264E cannot grow in M9 minimal medium with 0.04% glucose, 0.4% glycerol and 0.1% arabinose. However, In Figure 1A and Figure S1, the same strain is able to grow in the same media (with 0.2% arabinose) and generate B-galactosidase expression. How is the strain able to grow?

Reply: In Supplementary Fig. 3 we used ppGpp deficient **double mutant strain (Δ relA Δ spot)**, to determine the ppGpp production by the different RelA alleles, whereas in Fig. 1a and supplementary Fig 1 we used **Δ relA single mutant** strain to determine the effect of RelA on Hfq mediated basepairing (see legends and text).

4. In Figure 4 and S7, there is a poor reproducibility of the gels. In Figure 4D, very few Hfq multimers appear whereas most Hfq in Figure S7 is in a multimeric form (both in absence and presence of RyhB).

Reply: In this set of experiments, we investigated the effect of RelA on Hfq multimerization in the presence and in the absence of GGAGA RNA. The data presented in these experiments exhibit a RelA-mediated gradual increase in Hfq multimerization beginning from dimers to tetramers and

pentamers. This effect is dependent on the binding of RNAs with GGAGA which is clearly visible in both Fig. 4 and Supplementary Fig. 7. The multimerization experiments are highly susceptible to slight environmental and technical changes which can lead to a different starting point. However, in each of these experiments the controls are identical to each other; the multimerization pattern of Hfq with no RNA plus RelA is identical to that of Hfq with mutant RNA plus RelA. The gradual effect of RelA on Hfq multimerization is further confirmed by the bar-graph provided in case of both experiments. (Supplementary Fig. 6 and Supplementary Fig. 7).

5. Although the authors provide some more statistics, it is far from sufficient. No detailed explanation of what type and settings of the the t-test they are using (only “unpaired”). Also, the use of asterisks are very confusing, sometimes a “*” indicates a P-value < 0.5 (Figure 6) whereas it also can mean > 0.1 (Figure S1). The same goes for the other asterisks. Generally, asterisks should only be shown for results showing a statistical significance (P-value < 0.05). Otherwise, the results should be named ns (non significant).

Reply: As suggested, we replaced the asterisk with absolute P values in all figures. A two-tailed t-test was performed. This is now indicated in the appropriate legends.

6. Also, most of the experiments have only been performed once which is not sufficient since reproducibility is an issue (e.g. point 4 above for Figure 4D and Figure S7). This is especially important for “key” figures like Figure 5A where the authors claim that RelA facilitates time-wise progression of a RyhB:Hfq interaction.

Reply: **All the experiments were carried out several times.** For example, the data in Supplementary Fig. 7 is a repetition of Fig. 4. Supplementary Fig. 11 is a repetition of Supplementary Fig 10. In vivo ppGpp production (Fig. 6a) was carried multiple times (see standard deviation below the TLC). The repetition of Fig 6b is provided in Supplementary Fig. 13c. Supplementary Fig. 8 is a repetition of Figure 5a; the same experiment with another RNA. In these experiments we used wild type and mutated RelA protein, wild type and mutated RNAs and RNA competition as controls. The number of repetitions and standard deviations are indicated in the figure legends.

7. According to data in Figure S17, there is a very minute (close to none) effect of a RelAC289Y allele compared to RelAWT when determining the levels of sodA after Co-IP using RelA. However, as seen in Figure 2B and 3 the RelAC289Y mutant protein still binds much weaker to sodA compared to the WT protein. How can this be explained?

Reply: In Supplementary Fig. 17, the binding of sodA to RelA C289Y mutant is 1.6-fold lower compared to wild type. **However, in contrast to the review’s observation**, Fig. 2b and Fig. 3 show no binding of RelA C289Y mutant to neither RyhB nor SodA. The first is an in vivo experiment showing the binding levels after Co-IP the second presents in vitro binding data and hence the difference.

Minor comments:

8. Figure S3, why showing G251A, G251E and H354Y if not part of story?

Reply: In Supplementary Fig. 3 we present several newly constructed and previously known RelA mutants (G251A G251E and H354Y). The mutants relevant to the story are indicated in the legend to this Fig. We decided to leave the other previously known mutants as they may be of interest to investigators of RelA activity.

9. Why is the RelA:SodA complex much larger than RelA:RyhB complex in Figure 2C despite these complexes having a similar size in Figure 2B?

Reply: **Fig. 2c is a native gel**, hence the difference in the size of the complex, whereas **Fig. 2b is denaturing protein gel**.

10. Figure 5D, lanes 14-17 are still quite confusing, why is the shift lost in lane 16? Also, the brown asterisks sometimes show one complex or two complexes. In lane 7 for instance, a low Hfq concentration gives a large complex whereas such complex cannot be observed in lane 17. What does these results mean?

Reply: In this experiment we detect the formation of a mixed subunits hexamer that is based on minimal amount of WT Hfq bound to monomers of mutated Hfq. In the case of I30D, it is possible that formation of a mixed subunits hexamer, although stable, is less efficient when the mixture contains mostly I30D. However, when the ratio of I30D to WT is 50% we see a transition of the complex towards Hfq WT (compare lanes 6 and 16) while I30D complex is decreasing due to the inability of the low levels of I30D to join the mixed subunits hexamer complex. A further increase in the ratio of WT to I30D leads to the formation of a complex based mostly of WT Hfq (lane 17). Lane 3 (100 nM) exhibits 2 low and high complexes of WT Hfq, whereas lane 7 exhibits (5 nM Hfq and RelA) more of the second higher complex of Hfq and the beginning of the new third higher complex (probably because of RelA). Lane 17 contains more of the lower complex, forming only some of the higher complex, whereas the third high MW complex was not detected.

11. Figure 6B, despite claiming that in their answer to referees letter, the authors dis not repeat the experiment, or at least it is not written in the text.

Reply: The repetition of Fig 6b is provided in Supplementary Fig. 13c (this was mentions in the text, see page 13).

We thank the referees for their comments and hope that the revised manuscript is now acceptable for publication.

Sincerely,

Shoshy Altuvia, Prof.

REVIEWERS' COMMENTS

Reviewer #3 (Remarks to the Author):

Basu and co-workers have been able to clarify several points in their rebuttal and the paper could be suitable for publication if the reviewer could answer some yet unclear issues:

1. It is still almost impossible to get information of how many times an experiment has been repeated. In the rebuttal, the authors claim that "**All the experiments were carried out several times**" but I am still unable to get that information from most of the figure legends. The number of repetitions (n) of each experiment **must** be indicated for each Figure in the figure legends (e.g. Figure 1a. ... (n=5), b ... (n=?) and c ... (n=?)).
2. The data replicates of the different experiments **must** also be uploaded as source files together with the file shown in the manuscript. It is important to allow the reader to make his/her own judgement since some of the claims made by the authors are not very strong (e.g. the appearance of an extra band over time as indicated by the black arrow in Figure 5a).
3. I am still troubled by the lack of correspondence between in vivo and in vitro data for RelA binding to sodA. In vitro, RelAWT binds sodA strongly whereas RelAC289Y is unable to do so (Figures 2B and 3) whereas the in vivo difference between RelAWT and RelAC289Y binding to sodA is very marginal (1.6 times) (Figure S17). To me, this emphasizes that other factors might play a role and maybe more importantly, the danger of over-interpreting in vitro data. The authors should address these cautions in the discussion section.

February 10, 2021

Minju Ha, PhD
Associate Editor
Nature Communications
Re: Revised manuscript NCOMMS-20-32106B

Dear Dr. Ha,
Below is our response to the third referee last comments

1. It is still almost impossible to get information of how many times an experiment has been repeated. In the rebuttal, the authors claim that “**All the experiments were carried out several times**” but I am still unable to get that information from most of the figure legends. The number of repetitions (n) of each experiment **must** be indicated for each Figure in the figure legends (e.g. Figure 1a. ... (n=5), b ... (n=?) and c ... (n=?)).

We have added n=xx to each legend of figures presenting bar graphs with statistical representation. In addition, we added a statistics and reproducibility section in the main manuscript for all.

2. The data replicates of the different experiments **must** also be uploaded as source files together with the file shown in the manuscript. It is important to allow the reader to make his/her own judgement since some of the claims made by the authors are not very strong (e.g. the appearance of an extra band over time as indicated by the black arrow in Figure 5a).

All appropriate replicates have been uploaded or are presented in the supplementary file.

3. I am still troubled by the lack of correspondence between in vivo and in vitro data for RelA binding to sodA. In vitro, RelAWT binds sodA strongly whereas RelAC289Y is unable to do so (Figures 2B and 3) whereas the in vivo difference between RelAWT and RelAC289Y binding to sodA is very marginal (1.6 times) (Figure S17). To me, this emphasizes that other factors might play a role and maybe more importantly, the danger of over-interpreting in vitro data. The authors should address these cautions in the discussion section.

Even though the RelA mutant protein shows very little binding to sodA RNA in vivo, still no binding to RyhB can be detected under the same conditions, indicating that no regulation occurs. Therefore, we decided not to discuss this option in the discussion section.

Sincerely,

Shoshy Altuvia, Prof.